# A Comprehensive Review of Piezoelectric PVDF Polymer Fabrications and Characteristics

**DOI:** 10.3390/mi16040386

**Published:** 2025-03-28

**Authors:** Nadia Ahbab, Sidra Naz, Tian-Bing Xu, Shihai Zhang

**Affiliations:** 1Department of Mechanical Engineering and Aerospace, Old Dominion University, Norfolk, VA 23529, USA; nahba001@odu.edu (N.A.); snaz@odu.edu (S.N.); 2PolyK Technologies, LLC, 2124 Old Gatesburg Road, State College, PA 16803, USA; energy@polyktech.com

**Keywords:** piezoelectric, poly(vinylidene fluoride), material properties, fabrication, characterizations

## Abstract

Polyvinylidene fluoride (PVDF) polymer films, renowned for their exceptional piezoelectric, pyroelectric, and ferroelectric properties, offer a versatile platform for the development of cutting-edge micro-scale functional devices, enabling innovative applications ranging from energy harvesting and sensing to medical diagnostics and actuation. This paper presents an in-depth review of the material properties, fabrication methodologies, and characterization of PVDF films. Initially, a comprehensive description of the physical, mechanical, chemical, thermal, electrical, and electromechanical properties is provided. The unique combination of piezoelectric, pyroelectric, and ferroelectric properties, coupled with its excellent chemical resistance and mechanical strength, makes PVDF a highly valuable material for a wide range of applications. Subsequently, the fabrication techniques, phase transitions and their achievement methods, and copolymerization and composites employed to improve and optimize the PVDF properties were elaborated. Enhancing the phase transition in PVDF films, especially promoting the high-performance β-phase, can be achieved through various processing techniques, leading to significantly enhanced piezoelectric and pyroelectric properties, which are essential for diverse applications. This concludes the discussion of PVDF material characterization and its associated techniques for thermal, crystal structure, mechanical, electrical, ferroelectric, piezoelectric, electromechanical, and pyroelectric properties, which provide crucial insights into the material properties of PVDF films, directly impacting their performance in applications. By understanding these aspects, researchers and engineers can gain valuable insights into optimizing PVDF-based devices for various applications, including energy-harvesting, sensing, and biomedical devices, thereby driving advancements in these fields.

## 1. Introduction

Over the past half century, polymer-based piezoelectric materials [1] have gained significant interest in sensors [2], actuators [3], and energy harvesting [4]. Among these materials, polyvinylidene fluoride (PVDF) and its copolymers have emerged as highly valuable materials owing to their unique combination of properties [5]. PVDF, with the molecular formula (CH_2_-CF_2_)_n_, is a non-reactive thermoplastic known for its exceptional thermal, chemical, elastic, piezoelectric, and pyroelectric properties [6]. It is known to maintain its performance across a wide range of demanding environments, making it a versatile material for various advanced applications [7].

PVDF exhibits multiple crystalline phases, including the α-, β-, γ-, and δ-phases, each with distinct properties [8,9]. The α-phase is the most stable and is characterized by non-polar chains, making it less favorable for piezoelectric applications [10]. In contrast, the β-phase, which is polar and has an all-trans configuration, is highly desirable because of its superior piezoelectric properties [11,12]. The γ- [13,14] and δ-phases [15] offer intermediate characteristics, with the γ-phase being partially polar and the δ-phase being less common and less studied.

The discovery of piezoelectricity by the Curie brothers in 1880 laid the foundation for the development of numerous piezoelectric materials, including PVDF [16]. Although their discovery was not specific to PVDF, it opened the door to research various materials exhibiting this property. The piezoelectric properties of PVDF were first recognized in the late 1960s, with significant advancements soon following. In 1969, Kawai [17] demonstrated remarkable piezoelectric performance of PVDF, which was considered superior to that of other synthetic polymers at that time. This breakthrough spurred extensive research on PVDF and its copolymers, leading to the development of ultrasonic and electroacoustic actuators in the early 1970s [18].

In the 1990s and earlier 2000s, driven by the U.S. DoD ultrasonic transducer and robotics programs [19,20,21,22], as well as the NASA morphing program [23,24,25,26], the electroactive properties of PVDF and its various copolymers were studied intensively. These historical developments have significantly influenced current research trends, particularly the focus on enhancing the β-phase of PVDF [27,28,29], which is associated with its superior piezoelectric properties. The ability to control the crystalline phase of PVDF through precise processing techniques, such as mechanical stretching [30], annealing [31], and poling [32], has become a central aspect of modern PVDF research. These techniques are crucial for optimizing the piezoelectric performance of a material and expanding its application scope, which will be explored in detail in later sections.

PVDF’s copolymers, such as poly (vinylidene fluoride-co-hexafluoropropylene) (P(VDF-co-HFP)) [33], poly (vinylidene fluoride-co-trifluoroethylene) (P(VDF-co-TrFE)) [34], and poly (vinylidene fluoride-co-chlorotrifluoroethylene) [(P(VDF-co-CTFE)] [35], further enhance its functionality by offering tailored properties for specific uses. These materials combine flexibility, chemical inertness, and a wide frequency response range, making them ideal for demanding environments such as nuclear power [36], aerospace [37,38], automotive [39], biomedical [40], and industries [41].

In the last two decades, motivated by the U.S. nano initiative, many researchers have made great efforts to develop various PVDF-based nanocomposite materials [42]. Efforts have focused on developing nanocomposites [42] by introducing high dielectric constant fillers, such as ceramics, semiconductors, metal particles [43], graphene [44], and carbon nanotubes (CNTs) [45], into the PVDF matrix. These nanocomposites have significantly expanded the application range of PVDF, particularly in fields in which high performance and durability are crucial, such as aerospace [37], automotive [46], and construction [47]. Modification of dielectric properties through nanocomposites has enhanced applicability of PVDF in these sectors by improving its mechanical and electrical characteristics.

Despite having lower piezoelectricity compared with commonly used materials such as lead zirconate titanate (PZT) [48], PVDF and its copolymers offer several advantages, including light weight, low acoustic impedance, ease of fabrication, and chemical resistance. These benefits contribute to broad applicability of PVDF across diverse industries, including electronics [49], radio engineering, civil infrastructure healthy monitoring, architecture, and pharmaceuticals.

To ensure optimal performance of PVDF in a range of applications, it is essential to fully define its structural, electrical, and piezoelectric properties. Differential scanning calorimetry (DSC) [50], Raman spectroscopy [51], Fourier transform infrared spectroscopy (FTIR) [52], and X-ray diffraction (XRD) [53] are techniques commonly used to analyze the phase conformation and crystalline structure of PVDF. Scanning electron microscopy (SEM) [54] provides information on its shape and microstructure. The piezoelectric and dielectric properties are assessed using various methods, including dynamic [55] and quasi-static testing [56] for piezoelectric coefficients, dielectric constant tests, and piezo response force microscopy (PFM). Together, these techniques maximize the use of PVDF for complex functional applications. In addition, PVDF and its copolymer not only exhibit excellent ferroelectric properties for sensing, actuation, and transduction applications [19,20,23,25,57,58] but also are highly chemical resistant and relatively inert with very low surface energy that little can stick to [59,60,61]. Therefore, PVDF accounts for 54% of the PV back sheet market share [59]. The low surface energy means that PVDF materials can readily shed dirt and grime and easily clean off any kind of wet or dry dust/soil for transparency surface coating and other future applications.

Although many review papers on PVDF-based materials have been published [2,7,62,63,64,65,66,67,68], there is a gap in the comprehensive review of (i) various material properties for device engineering and developers, (ii) various characterization methods for materials scientists, engineers and physicists, and (iii) material property modifications for other researchers.

This paper is organized as follows: Section 2 highlights the major material properties of PVDF polymers, including their physical, mechanical, chemical, thermal, electrical, and electromechanical properties. Section 3 provides a detailed description of the fabrication methods and phase transition techniques used to enhance PVDF film, including stretching, annealing, and poling. Moreover, it provides details regarding the copolymerization of PVDF with various materials and their important composites. Section 4 provides PVDF film characterization techniques including FTIR, XRD, SEM, and DSC. Finally, a summary of the overall study is presented.

## 2. Material Properties

Material properties have been continuously investigated by researchers to understand and modify them for improved processing and functional performance. This section outlines the key properties of PVDF as shown in Figure 1, compares various processing techniques, and discusses their relative advantages and disadvantages in optimizing PVDF’s performance. Figures and tables are referenced to support these discussions.

### 2.1. Physical Properties

PVDF is an opaque resin with an extreme melting point (170–180 °C), allowing it to tolerate high temperatures without substantial deterioration. Controlling its optical properties, such as transparency, haze, and clarity, is crucial for specific applications. The fundamental causes of the limited visible-wavelength transmission and the significant mist in PVDF are its rough surfaces. It has a density of about 1.78 g/cm, which can increase to 1.97 g/cm^3^ with β-orientation of PVDF because of a greater level of crystallinity and higher packing density at β-phase. PVDF is characterized by its low weight and remarkable flexibility, which allows it to be readily molded into various shapes.

#### Crystaline Structure of PVDF

In most cases, the primary factors driving variations in the crystal structures of the molecular group are the internal configurations and the rotation of a single bond [30]. As a semi-crystalline piezoelectric polymer, the structure of PVDF is determined by the configuration of its chains (gauche and trans) and the orientation of the adjacent chains. PVDF can crystallize into nine distinct forms, grouped into five primary phases: alpha (α) [10], beta (β) [69,70], gamma (γ) [13,14], and delta (δ) [15], with the ε-phase [71]. Among these, the α-, β-, and γ-phases are the most important for practical applications, each offering distinct thermal, electrical, and mechanical properties. These crystal forms can be classified into four classes: I, II, III, and IV, corresponding to alpha (α) [10], beta (β) [69,70,72,73], gamma (γ) [13,14], and delta (δ) [15] and epsilon (ε) crystal phases, respectively [71]. Among these, the α-, β-, and γ-phases are the most significant, whereas the epsilon (ε-) and delta (δ-) phases are harder to isolate and are not typically produced through conventional methods. Each crystalline phase has distinct thermal, electrical, and elastic properties, as illustrated in Figure 2.

The α-phase is non-polar and electrically inactive, is directly obtained from the molten state, and is thermodynamically unstable. It features two anti-parallel chains in a trans-gauche-trans-gauche (TGTG) configuration [11,74]. In contrast, the β-phase is electroactive and exhibits strong ferroelectric and piezoelectric behavior [12]. Its all-trans planar zigzag conformation separates most fluorine atoms from hydrogen atoms, resulting in the maximum dipole instant for each unit cell among all the phases. This phase also generates a substantial polarization that occurs naturally because of dipole moment addition [39].

The dipole moments of the γ- and δ-phases are fewer and weaker than those of the β-phase, but they still possess polar unit cells. The γ-phase, which is an uncommon state between α and β, has a T3GT3G’ configuration with a higher trans fraction. A crucial difference between the α- and δ-phases lies in the orientation of the dipolar moments within the unit cell; in the δ-phase, they are parallel, whereas in the α-phase, they are antiparallel. The δ-phase, characterized by a TGTG structure, can be transformed into a β-phase material through the application of a high external electric field, ranging from 100 to 500 MV/m [10,11,12,13,14,15,16,17,18,19,20,21,22,23,24,25,26,27,28,29,30,31,32,33,34,35,36,37,38,39,40,41,42,43,44,45,46,47,48,49,50,51,52,53,54,55,56,57,58,59,60,61,62,63,64,65,66,67,68,69,70,71,72,73,74].

For many PVDF-based piezoelectric applications, it is essential to achieve a high electrical conductivity, dielectric permittivity, low dielectric loss, and high breakdown strength. Optimization of the β-phase significantly improves dielectric and piezoelectric properties of PVDF, making it highly effective for a range of applications.

### 2.2. Mechanical Properties

PVDF is a versatile polymer known for its excellent mechanical properties, which make it suitable for a wide range of applications. The mechanical properties of PVDF film includes tensile strength, elongation at break, Young’s modulus, yield strength, and impact strength. Tensile strength is the maximum stress of a material that can withstand tensile stress before breaking. Its units are pascals (Pa) and megapascals (MPa). A higher tensile strength indicates that the PVDFs have better breaking or tearing resistances. PVDF exhibits excellent tensile strength, meaning that it can withstand significant pulling forces before breaking. This property is crucial for applications that require high mechanical stress, such as structural components and protective coatings.

The elongation at break is the maximum amount of strain that a material can withstand before breaking. It is presented as a percentage (%). A higher elongation at break indicates better ductility and flexibility. PVDF also demonstrated good elongation at break, indicating its flexibility and ability to deform under stress without fracturing. This characteristic makes it suitable for applications involving bending or flexing. The Young’s modulus is another mechanical property that is a measurement of the stiffness of a material. This represents the ratio of stress to strain in the elastic region, where the units are pascals (Pa) or gigapascals (GPa). A higher Young’s modulus indicates stiffer material. Yield strength is the stress at which a material begins to deform plastically in units of pascals (Pa) or megapascals (MPa). A higher yield strength indicates a better resistance to plastic deformation.

Impact strength is the ability of a material to resist sudden impact forces in units of joules per meter (J/m) or foot-pound per inch (ft-lb/in). A higher impact strength indicates better resistance to shock and impact loads. PVDF possesses excellent impact resistance, making it capable of withstanding sudden shocks and impacts. This property is particularly valuable in applications in which the material may be subjected to accidental drops or collisions. PVDF exhibits good fatigue resistance, which means that it can withstand repeated cycles of stress without experiencing significant degradation. This property is essential for applications that involve continuous or cyclic loading. PVDF also demonstrates excellent creep resistance, indicating its ability to maintain its shape and dimensions over time under a constant load. This property is crucial for applications that require long-term dimensional stability. In addition, PVDF has a high melting point and excellent thermal stability, allowing it to withstand high temperatures without significant degradation. This property makes it suitable for applications in harsh environments. Specific equations for calculating these mechanical properties can be found in [75,76]. The factors that influence the mechanical properties of PVDF include the molecular weight, crystallinity, processing techniques, and post-processing treatments. Higher molecular weight and crystallinity improve mechanical properties, such as tensile strength and modulus. Processing techniques such as extrusion and molding can influence the final properties. Postprocessing techniques such as stretching and annealing can further enhance mechanical properties. PVDF is frequently employed to coat the outside surfaces of buildings because it exhibits little to no degradation in mechanical quality over several decades of outdoor use. Among traditional crystalline polymers, PVDF is particularly notable for its mechanical properties. It has the greatest impact strength and is ranked second after polyoxymethylene based on the degree of hardness, tensile strength, compressive stress, and flexural stress stiffness [64]. The various properties of PVDF are provided in Table 1.

### 2.3. Chemical Properties

The solubility of a substance is affected by its chemical and physical characteristics, as well as by external variables such as temperature, pressure, and the presence of other chemicals in the solution. PVDF, an organic polymer, follows the “like-dissolves-like” rule, with polarity playing a key role. PVDF is soluble in a few organic solvents, such as dimethylformamide (DMF), dimethyl sulfoxide (DMSO), N-methyl pyrrolidone (NMP), and dimethylacetamide (DMAc), but is generally insoluble in aliphatic compounds, aromatic compounds, chlorine solutions, alcohols, strong acids, halogens, and basic solutions. The PVDF copolymers tend to be slightly more soluble because of their lower crystallinity [36]. PVDF is recognized for its excellent chemical resistance, which makes it suitable for harsh environments. However, they react negatively with ketones, esters, and other strong alkalis [79]. Over time, the properties of PVDF were weakened by exposure to strong alkaline solutions [80].

For example, NaOH can cause discoloration and brittleness in PVDF membranes. The degradation mechanism involves dehydrofluorination, which leads to the formation of carbon–carbon double bonds and weakening of the polymer chain [81]. Studies have shown that the addition of stress and strain accelerates degradation. When exposed to NaOH, PVDF’s mechanical strength diminishes, as confirmed by various tests and spectroscopic analyses [82]. In addition to NaOH, PVDF stability was tested using sodium hypochlorite (NaOCl) and potassium hydroxide (KOH) [83]. Overall, PVDF demonstrates outstanding chemical stability and durability; however, careful consideration is required when using strong alkaline solutions to prevent degradation. Table 2 lists the chemical properties of PVDF.

### 2.4. Thermal Properties

The thermal stability of polymers refers to their ability to resist heat and maintain properties such as strength, durability, and flexibility. Two key parameters influence the thermal stability of PVDF: the glass transition temperature (T_g_) and the melting temperature (T_m_). PVDF has a T_g_ between −20 °C and 60 °C, making it a rubbery polymer above 0 °C. Melting temperatures are between 160 °C and 190 °C, which is dependent on processing and phase conditions. Every phase of PVDF has comparable melting temperatures, with slight variations in T_m_ indicating various crystalline phases. PVDF has excellent thermal stability, with temperatures ranging from −30 to 170 °C. The temperature of thermal degradation also remained constant, ranging from 400 to 450 °C, regardless of the crystalline phase, quantity of crystallinity, and manufacturing technique. Most data suggest that the Curie temperature (T_c_) of PVDF, where it loses spontaneous polarization, is between 195 °C and 197 °C. The low thermal conductivity of PVDF makes it ideal for insulation, and its low coefficient of linear thermal expansion ensures minimal dimensional changes at different temperatures [84].

### 2.5. The Electroactive Properties of the PVDF

PVDF is notable for its exceptional piezoelectricity, which refers to its capability to produce an electric charge when a mechanical stress or force is applied, as well as generating mechanical stress whenever an electrical field is applied. Similar to other ferroelectric materials, PVDF follows the Heckman diagram representing the interrelationship between the mechanical, thermal, and electrical properties of materials [85].

#### 2.5.1. Ferroelectric Effect

The ferroelectricity of PVDF is caused by dipole moments, inside its molecular structure, which are mainly induced by strongly electronegative fluorine atoms as shown in the β-phase in Figure 2b. These dipoles align in response to an electric field, allowing the material to create a hysteresis loop, which is essential for the ferroelectric activity. PVDF’s ferroelectric properties, discovered in 1974 [86], enhance its piezoelectricity by allowing it to maintain polarization long after the electric field is removed. However, as-manufactured PVDF films, particularly those formed from melt or solution, lack ferroelectricity because their dipole moments cancel due to their crystal structure. Additional approaches, such as mechanical stretching or the use of copolymers such as P(VDF-TrFE), are necessary to induce ferroelectricity [87]. These techniques enhance the dipole alignment, resulting in ferroelectric behavior. PVDF demonstrate a shift in the phase and transition temperature with an applied electric field, like ferromagnetic materials [88]. When heated over the *T*_*c*_, the material transitions from the ferroelectric β-phase to the paraelectric α-phase. Below this temperature, PVDF maintains spontaneous polarization [89].

##### Dielectric Properties

The energy dissipation in PVDF is also known as dielectric loss (tanδ) [20]. This describes how energy is lost in dielectric materials under an electric field. For PVDF, the dielectric loss is influenced by frequency and nanofiller presence, typically ranging from 0.04 to 0.11 depending on the sample [90].

##### Piezoelectric Effect

The expression “piezoelectricity” refers to a property of a material to convert intrinsic elasticity into dielectric energy if it is exposed to an external pressure or stress [91,92,93]. In PVDF, this effect can be quantified through several important constants such as the piezoelectric strain constant (d_ij_), piezoelectric voltage constant (g_ij_), and electromechanical coupling coefficient (k_ij_) [94], which characterize the efficiency and sensitivity of the piezoelectric response of PVDF to mechanical stress and deformation. The piezoelectric charge constant (d_ij_) is a key parameter that describes the relationship between mechanical stress and electric polarization for device applications.

Direct Piezoelectric Effect: A piezoelectric material produces an electric charge on its surface when a mechanical force is applied. This charge can be gathered and used for several devices, including sensors, transducers, and energy harvesters [95,96,97].

Inverse Piezoelectric Effect: A piezoelectric material undergoes deformation or shape changes when an electric field is applied. The inverse piezoelectric effect serves as the basis for an electromechanical actuator for electrodeformation. Utilizing the reaction of the material to an applied voltage to provide precise control has been exploited in various piezoelectric actuators [98,99,100].

Piezoelectric Strain Constant (d_ij_): 

The piezoelectric coefficients (*d*_ij_) form a matrix that expresses the behavior of the material in different directions. The matrix form for the piezoelectric coefficients is represented in reference [101], and the relationship for PVDF is described in reference [102].

For PVDF, the most commonly measured coefficient is d_33_, which describes the charge generated along the thickness of the material when subjected to stress in the same direction [12]. Another typical piezoelectric coefficient is the transverse mode (d_31_), in which mechanical stress is applied at a right angle to the polarization direction.

Piezoelectric Voltage Constant (g_ij_): The piezoelectric voltage constant (g) represents the electric field generated per unit of applied mechanical stress [103]. This is crucial for sensing and energy-harvesting applications. The dielectric constant ε of piezoelectric materials is another key parameter that determines the manner in which a material stores electrical energy in the presence of an electric field.

Electromechanical Coupling Coefficient (k_ij_): The electromechanical coupling coefficient is used to directly assess the efficiency of energy conversion in a material [104].

#### 2.5.2. Pyroelectric Effect

While the piezoelectric effect involves generating charge in response to applied mechanical stress, the pyroelectric effect involves generating charge owing to changes in temperature [105,106]. This effect arises from the spontaneous polarization of the molecular structure of PVDF, primarily in its β-phase, where dipoles align along the polymer chains. This property is particularly useful in applications such as infrared sensors, thermal detectors, and energy-harvesting devices [107]. Pyroelectricity in PVDF is often enhanced by processes such as stretching and poling, which align the dipoles more effectively, increasing the sensitivity of the material to temperature variations.

## 3. Enhancement of PVDF Properties Through Optimized Preparation Methods

The PVDF structure directly influences its characteristics. For example, in certain fields, PVDF is used for different purposes, such as films, fibers, or complex shapes with different thicknesses to meet the needs of different industries. The dielectric permittivity and electrical conductivity of PVDF are known to increase with increasing β-phase content. To improve the other properties of PVDF, it can be formed into two fibers or layers. Moreover, PVDF with piezoelectric characteristics can be achieved through straightforward material modifications, such as phase transitions [108], or by using advanced processing technologies [66]. In addition, achieving the desired properties requires the use of PVDF copolymers [109]. Figure 3 shows different methods for improving the PVDF properties. Many researchers have also improved various properties of PVDF, especially piezoelectricity, by blending it with polymers or adding fillers [110]. In addition, different grades of PVDF, such as high-molecular-weight PVDF or low-molecular-weight PVDF, play a crucial role in achieving certain goals.

### 3.1. PVDF Film Fabrication Methods

PVDF films are the most popular structures for use in various device applications because (i) films easily form piezoelectric status with relatively lower force stretching to form β-phase and lower voltage to pole to permanent polarization and (ii) films make functional devices for sensors, actuators, transducers, and energy harvesting with (a) more efficiency, (b) lower cost of operations, and (c) lower cost of fabrications. The manufacturing process plays a crucial role in shaping PVDF’s diverse features, including its superior dielectric and piezoelectric properties. PVDF can be fabricated using a various technique to create α- or β-phase materials. The hot pressing method, 3D printing, spin coating, electrospinning, and solution casting are examples of common techniques. These methods frequently result in Langmuir–Blodgett films, which are nanostructured solid films formed at the liquid–gas interface. However, solution casting techniques, such as electrospinning, solvent evaporation, and spin coating, have received increased attention because of the need for films with nanoscale thicknesses in numerous applications.

#### 3.1.1. Solution Casting

Solution casting is a common method for producing PVDF and its copolymer-based films. This method can be used to obtain excellent films with better mechanical, optical, and physical characteristics through unformed in-plane distributions. Certain organic solvents, including DMF [111], methyl ethyl ketone (MEK) [112], dimethyl sulfoxide (DMSO) [113], acetone [114], N,N-dimethylacetamide (DMAc) [115], and 1,3-dioxolane (DXL) [116], can dissolve PVDF and its copolymers.

The dissolution process usually takes place at room temperature but sometimes requires a higher temperature (<70 °C) [117]. One popular technique for enhancing dissolution is ultrasonication-aided mixing technology. In most cases, the ultrasonication operation time is less than 30 min, but it can vary case by case and range from a few minutes to three hours [115]. Stirring, as compared with ultrasonication, is typically performed for more than 30 min. After evaporation, the fully mixed PVDF/organic solution was placed over an aluminum (Al) substrate or glass to create the first films [115]. Evaporation can occur in vacuum chambers or in the air, at temperatures ranging between 50 °C and 120 °C [117].

#### 3.1.2. Solution Coating

Solution coating is another commonly employed method for creating composite membranes. This method creates composite membranes by applying a thin layer of the chosen polymer solution over a porous substrate. Because a selective coating layer can regulate them, care should be taken beforehand to ensure that the substrate’s degree of porosity is larger than the membrane resistance. A low-boiling-point solvent, which is insoluble in the coating solvent and can reduce invasion, was applied to the substrate prior to the coating process. The solvent was evaporated to create a coated membrane. The primary benefit of solution coating is its ease of fabrication, which can produce homogenous coatings over large areas at low temperatures [118,119,120,121].

#### 3.1.3. Spin Coating

The spin-coating method was used to prepare thin film, which is a cost-effective and time-efficient method for processing uniform polymer films from a dilute solution on a planar substrate [31]. This method is suitable for producing smooth polymer coatings for microelectronic applications. Researchers have successfully prepared nanoscale films of crystalline PVDF using this technique. Key factors, such as humidity and rotation speed, can be used to measure the surface roughness and phase content of thin films [122]. Additionally, the thickness of the films depends on the properties of the solvent solution and the spin speed. However, the thin films were pure non-poled alpha phases of PVDF, necessitating further polarization of the piezoelectric response [109,123]. Spin coatings have more advantages when integrated with silicon-wafer-based microelectrocnics or micromachines.

#### 3.1.4. Hot Pressing Method

The fabrication of PVDF films using hot pressing began with the preparation of the PVDF powder or pellets. The material was melted under controlled conditions, typically at temperatures ranging from 150 °C to over 200 °C [124], ensuring complete melting without decomposition. High pressures, often between 5 and 40 MPa [125], were applied to compress the melting power to enable the formation of a continuous film with eliminated voids and enhanced density. Following the application of heat and pressure, the films were cooled in a controlled manner, solidifying the polymer into the desired crystalline phases. Additional treatments, such as stretching or electrical poling, can be applied to optimize the piezoelectric properties of the material [124].

Hot pressing is recognized for its ability to improve the crystalline phases of PVDF, especially the transition from the α-phase to the β-phase, which is critical for piezoelectric applications [126]. This technique is particularly advantageous because of its ability to produce dense films with high β-phase content, superior dielectric properties, and improved mechanical performance. For instance, studies have demonstrated that varying the hot press temperature influences the crystallite size of the PVDF films. At temperatures around 170 °C, the crystallite size significantly increased, enhancing the β-phase content and resulting in films with superior piezoelectric and dielectric properties. Specifically, crystallite size has been observed to grow from 7.2 nm to 20.54 nm with increased pressing temperature [127].

The hot pressed PVDF films also exhibited exceptional dielectric properties. Films processed at 150 °C have been reported to achieve high energy densities (19.24 J/cm^3^) coupled with an efficiency of 68.99%. Additionally, their breakdown strength of 1189 kV/cm and dielectric loss as low as 0.04 make them highly suitable for high-performance energy storage applications [128]. The temperature and pressure applied during hot pressing play critical roles in determining the crystallinity and phase composition of the films. Rapid cooling post-pressing has been shown to promote β-phase retention, thereby further enhancing the functional properties of the material. Comparative studies highlight the superiority of hot pressing over other fabrication methods, such as solvent casting. Hot pressing provides more controlled thermal and mechanical treatment, yielding films with a higher β-phase content and fewer defects. For instance, the uniform application of pressure and heat during hot pressing results in films with enhanced thermal stability and superior dielectric and mechanical properties compared with films produced using other techniques [129].

Massive high-quality commercial piezoelectric films are fabricated with a hot-pressing-based comprehensive production line, which is integrated with hot pressing to form a film, mechanical stretching to transfer to the high-yield β-phase, and post-annealing to increase crystallinity. With the large-scale commercial production lines, it is relatively easy to control the temperature and stretching uniformity to obtain high-quality piezoelectric PVDF films. However, it is very challenging to obtain high-quality films using the hot pressing method at a small lab scale. Therefore, the solution casting method is more convenient for use in academic laboratories.

#### 3.1.5. Electrospinning

Electrospinning is a popular technique for creating PVDF fiber-based films, particularly ultrathin films. This method involves solution preparation, electrostatic field application, jet formation, jet stretching, and solvent evaporation. Electrospinning of PVDF films is a multifaceted approach that includes the application of a high voltage to a polymer solution, resulting in the formation of a thin jet that dries into nanofibers upon interaction with a collector. This technique utilizes the electrostatic forces produced between the charged polymer solution and the grounded collector, leading to the creation of fibers with diameters ranging from nanometers to micrometers. The fundamental concept is in the interaction of surface tension, electrostatic repulsion, and solvent evaporation, which together dictate the jet path and fiber development.

Electrospinning has advantages over alternative techniques because is causes PVDF molecular chains to elongate uniaxially along the fiber axis, resulting in the formation of the β-phase [130]. PVDF films can exhibit piezoelectricity without post-poling because polymer jet elongation and whipping allow poling to occur during the electrospinning process. Electric forces can complete the poling and stretching processes in electrospinning [7]. PVDF membranes with appropriate crystallinity can be created by applying heat treatment during or after electrospinning. Cross-linked membranes with ideal mechanical characteristics can be fabricated by adjusting the heating procedure during or after the electrospinning [131]. Research has also explored the combination of hot pressing and electrospinning, with final products obtained by compressing dried electrospinning PVDF/SMG membranes [6]. The primary variables in the electrospinning process include the applied voltage, needle–collector distance, and flow rate [132].

#### 3.1.6. 3D Printing

Additive manufacturing, or 3D printing, is another important method for the preparation of PVDF-based piezoelectric materials, as shown in Figure 4. This technique allows the creation of precise shapes through layer-by-layer deposition. Fused deposition modeling (FDM) is one of the various methods owing to its affordability and user-friendliness [133]. PVDF samples of various forms can be produced using FDM by increasing the nozzle temperature beyond the melting point of the material. Although FDM can induce some β-phase formation through stress, it is not sufficient to generate a high β-phase content (>50%) [134].

Researchers have aimed to enhance the β-phase content by optimizing the printing parameters, utilizing P(VDF-TrFE) with a high ferroelectric phase content, and incorporating additives. After achieving high β-phase content, post-processing poling is necessary to align the dipoles for macroscopic piezoelectricity; for example, an “integrated 3D printing and corona poling process” (IPC) was developed by Sebastian et al. [135], enhancing the β-phase content to 56.8%. Similarly, the electric-poling-assisted additive manufacturing (EPAM) process, developed by Lee and Tarbutton [136], employs mechanical stretching and electrical poling to convert α-phase PVDF into the β-phase. Although these methods increase the β-phase content, achieving improved piezoelectric properties remains a challenge. Substituting PVDF with P(VDF-TrFE) proves effective, as demonstrated by [137], who achieved 85.8% crystallinity and a d_33_ value of −18 pC N^−1^ with in situ polarized P(VDF-TrFE). Multi-layered or specially shaped designs can further enhance piezoelectricity, as demonstrated by Yuan et al., whose six-layer P(VDF-TrFE) device exhibited a high d_33_ of −130 pC N^−1^ [138]. In conclusion, FDM and other 3D printing techniques hold great promise for producing high-precision, cost-effective PVDF-based piezoelectric materials, with ongoing research aimed at increasing β-phase content. Table 3 lists the different fabrication methods, along with their advantages and disadvantages.

There have also been reports of other techniques, such as tape casting, template, and phase separation; however, because of drawbacks such as irregularity, synthesis process costs, complexity, and forms, they are not covered here [139].

### 3.2. Phase Transition and Achievement Methods

Stretching, annealing, and poling of the polymer in strong electric fields are common phase transition processes in PVDF crystals, as shown in Figure 5. Processing the solution with polar solvents at a crystallization temperature below 70 °C provided the β-phase, whereas processing at temperatures over 110 °C mostly produced the α-phase. Recognizing and controlling the phase transition is essential for adjusting the properties of PVDF for specific applications. Additionally, these methods work well for producing films with thicknesses of several micrometers. However, a major drawback of the phase transition is that it rarely occurs, leaving approximately 20% of the α-phase in the material after adapting to the β-phase PVDF [7].

#### 3.2.1. Stretching

Kawai first described the possibility of a piezoelectric effect on a stretched and poled PVDF film in early 1969. Subsequently, numerous studies have investigated the effect of stretching criteria on the piezoelectricity of PVDF. These criteria include the temperature stretching, stretching rate, stretching ratio, and stretching direction, which need to be considered carefully [30]. It is generally known that ferroelectric β-phase crystals may grow in PVDF under the stretching circumstances, greatly boosting the piezoelectricity of the material when attempting to follow electric poling. Without electric poling, even the 100% β-phase PVDF films exhibited no piezoelectric activity [141]. The influence of stretching temperature on the composition of the β-phase in PVDF was examined in most investigations using FTIR [124]. One of the key factors influencing the crystalline phase of PVDF is stretching temperature. The ideal temperature range to cause β crystals to form varies depending on the original PVDF films’ composition and processing parameters. Some highlighted studies based on stretching temperature are reported in these articles [142,143].

Figure 6 displays the scattering intensity profiles from two-dimensional wide-angle X-ray (2D-WAXS) patterns in the meridional direction based on the strain at 60 °C and 140 °C [142]. According to published data, the profiles of PVDF prior to stretching at both temperatures exhibit three distinctive peaks that are attributed to the (100), (020), and (110) diffractions of a-form PVDF. Without stretching at various temperatures, it is deduced that the a-form crystal of the PVDF sample is predominant. The findings show that the scattering curves evolve similarly during deformation at both temperatures.

Another crucial factor influencing the formation of the β-phase in PVDF, as seen in Figure 7, is the stretching ratio. According to certain studies, it even promotes the production of β crystals in PVDF more effectively than temperature [124]. In a PVDF film, the *α*-phase eventually changed into the β-phase by increasing the stretching ratio [144]. The reason for this was that the increased stretching ratio may aid in aligning the polymer chains, favoring crystallization into the all-trans conformation’s more compact β-phase.

Figure 8(a_1_) shows the crystalline phase conversion in the PVDF films as determined by FTIR spectroscopy. The PVDF films showed a phase transformation from the α- to β-phase as the stretch ratio increased [145]. This was demonstrated by the α characteristic absorption bands of the phase at 764 and 975 cm^−1^ becoming weaker and the characteristic absorption band of the β-phase at 840 cm^−1^ becoming stronger. The additional phase content quantification is shown in Figure 8(a_2_), which demonstrates that when the films were stretched to a ratio of 5, the β-phase content grew significantly and subsequently declined somewhat. R = 3 yielded the highest β-phase content, 88.18%. Figure 8(b_1_) displays the DSC thermographs of the stretched PVDF films as a function of temperature, whereas Figure 8(b_2_) displays the matching melting temperatures (T_m_) and crystallinity values determined from the DSC. The crystalline structural characterization of the PVDF films utilizing 1D wide-angle X-ray diffraction (WAXD) profiles is shown in Figure 8(c_1_). Figure 8(c_2_) provides a detailed analysis of the change tendency of 2θ and the crystallite size derived from the (110)/(200)β-PVDF peaks with increasing stretch ratio. The results show that when the stretch ratio increased from R = 1 to R = 5, the crystallite size reduced from 6.51 nm to 3.86 nm, and 2θ decreased from 21.03° to 20.63° [145].

The efficiency of the phase transition between the nonpolar *α*-phase toward the polar β-phase is also significantly influenced by the stretching direction. Biaxial stretching can provide homogeneous thicknesses for PVDF films while enhancing isotropic piezoelectricity in comparison with uniaxial stretching. In contrast to uniaxially oriented films, the polarized biaxially oriented poly(vinylidene fluoride) (BOPVDF) films demonstrated balanced piezoelectric activity in the film plane, which in turn led to a greater piezoelectric coefficient, according to the research by Mohammadi et al. [146]. Biaxial stretching is simpler for generating β crystals than uniaxial stretching, according to Ting et al. [104]. Figure 9 shows the experimental setup, which consists of a two-direction planar stretching machine, which measured 1100 × 1000 × 250 mm and stretched the PVDF thin films uniaxially and biaxially at varying temperatures and stretch ratios. Figure 10 shows the test setup for the measurement of the piezoelectric coefficient of biaxial stretching of PVDF films [104].

The FTIR findings of uniaxial stretching with a stretch ratio of R = 5 and biaxial stretching with stretch ratios of R = (2 × 2), R = (3 × 3), and R = (4 × 4) at various temperatures are shown in Figure 11. The arrows indicate the absorption band characteristics of each phase in the FTIR spectra, making it simple to identify the α- and β-phases. The FTIR spectra of uniaxially stretched PVDF samples are shown in Figure 11a. The FTIR spectra of the biaxially stretched PVDF samples are shown in Figure 11b–d. It is evident that several α- and β-phase absorption peaks in biaxial stretching differ from those in uniaxial stretching. The absorption peaks of the phase at 763 cm^−1^ increased as the temperature increased in both uniaxial and biaxial stretching, whereas the β-phase at 840 cm^−1^ decreased, suggesting that there was less transition from the α-phase to the β-phase [104].

In conclusion, mechanical stretching is a viable method for causing PVDF films to undergo a phase transition from α to β. Piezoelectric coefficients often increase with the amount of the β-phase. It is very desirable to appropriately adjust the stretching parameters in order to enhance the content of β crystals [12].

#### 3.2.2. Annealing

The piezoelectric properties of PVDF-based polymers are frequently enhanced by thermal annealing [122]. Two important factors that affect piezoelectricity during annealing are temperature and time [147,148]. The annealing temperature is usually adjusted between the T_c_ and the melting temperature (T_m_) when polymers exhibit considerable chain mobility in their paraelectric phase [149]. Because T_c_ is greater than T_m_ for PVDF at ambient pressure, annealing is a common method to alter the crystalline structure of P(VDF-TrFE) films. The spherical thickness and degree of crystallinity were often improved by longer annealing times [150,151]. Owing to the enhanced molecular mobility that causes dipole depolarization, annealing can occasionally result in a decrease in piezoelectric coefficients [152]. Annealing can realign and reorient crystalline phases in electrospun fibers, increasing their piezoelectric coefficient and overall dipole moment [148,153]. Furthermore, the cooling rate in the post-annealing process plays a significant role in the β-phase formation. When the annealed PVDF is cooled rapidly, the amount of β-phase increases.

### 3.3. Poling

In semicrystalline materials, poling is essential for realigning molecular dipoles and reorienting crystallites to maximize the piezoelectric effect. By inducing β-phase crystal formation in PVDF-based polymers, this technique achieves macroscopic polarization and piezoelectricity by aligning ferroelectric domains [154]. The most popular methods are contact poling [155], in situ polarization [156], and corona poling [157]. Using asymmetric electrodes at high voltages, corona poling produces a uniform electric field that increases the content of the β-phase crystals [158]. However, corona poling can introduce conductive charge carriers and reduce crystallinity [12]. Contact poling, often used for preparing piezoelectric PVDF films, is influenced by parameters such as the electric field, time, and temperature [159]. Higher electric fields induce a higher β-phase content, but exceeding the breakdown strength of PVDF can cause early failure. DC (~10^6^ V/m) poling at elevated temperatures (~393 K) is preferred over AC because of its higher breakdown strength [160,161]. The film thickness must be controlled to achieve a high poling field [12]. The poling time should be optimized to avoid structural defects [157]. The structure between T_c_ and T_m_ is effectively changed by the ideal poling temperature, which is typically between 20 and 120 °C. However, electrical poling has certain practical and cost-effective limitations.

Figure 12 shows a schematic of the in situ polarization used for the PVDF films [156]. Figure 13 shows the piezoelectric coefficients of the PVDF films at various polarization voltages [156]. Figure 14 shows the XRD graphs of the PVDF copolymer obtained by the in situ polarization method after poling, where the A, B, and C curves represent the poled, annealed, and dried processes, respectively [9]. Figure 15 shows an example of the corona poling schematic setup. Figure 16 presents examples of the XRD patterns and FTIR spectra of PVDF obtained after the corona polarization process. Figure 16a shows the XRD patterns, and Figure 16b shows the FTIR spectra of PVDF, where the (a) curve presents the received PVDF films, the (b) curve shows the stretched PVDF, and the (c) curve indicates a stretched and poled PVDF [157]. Table 4 lists the crystalline phase transformation methods of the PVDF polymer, along with their advantages and disadvantages.

### 3.4. Copolymerization of PVDF

Copolymerization allows the modulation of intramolecular and intermolecular forces, enabling changes in properties such as the melting point, glass transition temperature, crystallinity, stability, elasticity, permeability, and chemical reactivity over a wide temperature range. An electroactive phase can be achieved by combining PVDF with trifluoroethylene (TrFE), hexafluoropropylene (HEP), chlorotrifluoroethylene (CTFE), or other polymers modified for specific applications [33,164]. Table 5 lists the advantages and applications of PVDF copolymers. Figure 17 shows the various PVDF copolymers chemical compositions.

#### 3.4.1. P(VDF-co-HFP)

One widely used PVDF copolymer is poly (vinylidene fluoride-hexafluoropropylene), which is formed by polymerizing vinylidene fluoride (VDF) with hexafluoropropylene (HFP). The incorporation of bulky CF_3_ groups from HFP significantly reduces the crystallinity, adversely affecting the piezoelectric properties of the materials. However, in comparison with pure PVDF, flexibility increases significantly and can also enhance the ferroelectric properties [165]. The addition of HFP results in an increase in the fluorine content, which also improves the hydrophobicity [166]. Although its pyroelectric coefficient is higher, the piezoelectric coefficient (d_31_) is comparable to that of PVDF [167]. The high piezoelectric response makes P(VDF-co-HFP) suitable for applications in flexible energy harvesters, converting mechanical energy from vibrations and human motion into electrical energy. Owing to its excellent ferroelectric properties, it is used in sensors and actuators, which require materials with high sensitivity and flexibility.

#### 3.4.2. P(VDF-co-TrFE)

Poly(vinylidene fluoride-co-trifluoroethylene) (PVDF-TrFE) is a PVDF copolymer synthesized by polymerizing VDF and TrFE. The distinguishing features between PVDF and P(VDF-TrFE) are listed in Table 5. PVDF-TrFE has higher crystallinity and preferred orientation, enhancing the electromechanical coupling factor [168]. The addition of TrFE with extra F^−^ facilitated the formation of the β-phase, allowing polarization without stretching. PVDF-TrFE exhibited a lower T_c_ due to the reduced interactions between the units and dipole moments [116]. Its characteristic peak shifts to 19.9° and exhibits a rod-like grain shape in the FTIR spectrum [34]. The introduction of TrFE improves the flexibility and mechanical strength of the polymer, making it suitable for dynamic applications. It is used in the fabrication of flexible electronic devices, including wearable sensors and flexible displays, owing to its high dielectric and piezoelectric properties.

P(VDF-TrFE) is an upgraded variant of PVDF that exhibits markedly improved piezoelectric and ferroelectric characteristics. The unstretched and unpoled PVDF-TrFE exhibits a strong piezoelectric effect, which significantly reduces the device fabrication cost for sensing applications. For example, PVDF-TrFE can be directly spin-coated onto a silicon wafer to form sensors. However, pure PVDF cannot form sensors directly without stretching and poling. The piezoelectric constants of PVDF-TrFE are significantly higher than those of PVDF films under the same stretching and poling conditions. These enhancements provide it a more adaptable and useful material for many applications, especially those dependent on robust electromechanical coupling. Table 5 shows the distinguishing features between PVDF and P(VDF-TrFE).

#### 3.4.3. P(VDF-co-CTFE)

High flexibility, electromechanical response, elongation, and superior cold resistance are characteristics of P(VDF-co-CTFE), a semi-crystalline polymer with a lower degree of crystallinity than that of PVDF [35]. Its higher piezoelectric constant and the presence of Cl and F enhance safety and improve the mechanical properties [169]. Its C–Cl bond offers an active site for modification, making it a promising membrane material. Additionally, CTFE enhances the thermal stability and ferroelectric properties of the polymer, making it more suitable for high-temperature applications and ideal for sensors and devices that operate in high-temperature environments, such as industrial monitoring equipment. Its excellent piezoelectric properties make it suitable for biomedical devices such as medical implants and diagnostic equipment, which require precise sensing and actuation [170]. The main advantages and highlighted applications of PVDF copolymers are listed in Table 6.

### 3.5. PVDF Composites

Due to the intriguing properties of PVDF, there is a great deal of interest in this material, which has led to its combination with certain fillers to create high-performance, multifunctional PVDF-based composites with unique morphologies and physicochemical characteristics. PVDF-based composites are the outcome of combining one or two different fillers with complimentary qualities to enhance certain particular traits or to introduce new ones, such as magnetic or electrical conductivity.

#### 3.5.1. Fillers

PVDF composites with more than 30 distinct fillers have been reported [171]. To decrease the amount of filler and improve the functionality, it has recently become popular to build PVDF composites by adding several fillers with complimentary qualities [171]. Magnetic nanoparticles (CoFe_2_O_4_ or Fe_3_O_4_) [172]; carbon nanotubes (CNTs) [173]; and ceramic particles such as barium titanate (BaTiO_3_) [174], zinc oxide (ZnO) [175,176], titanium dioxide (TiO_2_) [177], zeolite, and clays [178] are among the most representative and frequently used fillers in PVDF and its copolymer-based composites. Figure 18 shows the effect of various nanofillers on PVDF properties and their applications. These fillers have also encouraged the development of a wide range of multifunctional composite materials. PVDF composites are intriguing because their final properties may be precisely controlled by appropriate filler size, shape, and content selection, as well as through filler and polymer interaction, dispersion, interface, and processing conditions [179].

#### 3.5.2. Enhanced Piezoelectric Properties by PVDF Blend with Polymers

PVDF and its copolymers are often blended with various polymers to enhance their properties. The classifications of thermoplastics blending are shown in Figure 19. Common blends include poly(methyl methacrylate) (PMMA) [180], poly(o-methoxyaniline) (POMA), poly(aniline) (PANI) [181], poly(L-lactic acid) (PLLA) [182], poly(ethylene terephthalate) (PET) [183], poly(vinyl chloride) (PVC) [184], poly(ethylene oxide) (PEO) [185], poly(vinyl alcohol) (PVA) [186], poly(carbonate) (PC) [187], and poly(amide 11) (PA11) [188]. These blends have better processability, induce certain crystalline phases, and adjust electrical and optical properties.

The PVDF/PMMA blend stands out, as it promotes β-phase formation and enhances the piezoelectric effect [190]. This blend has been used in optical applications [191], Li-ion battery separators [192], wettability switching [193], pyroelectric applications, and protective coatings. Another notable blind/mixture is PVDF/PLLA, which combines two piezoelectric polymers for energy-harvesting mechanisms. PVDF is also blended with conductive polymers, such as polypyrrole (PPy) [194], PANI [195], and PEDOT [196], making it ideal for use in actuators, sensors, and medical applications. To improve the dielectric flexibility and strength, PVDF was blended with poly(VDF-ter-TrFE-ter-CFE), which is suitable for dielectrics and energy storage owing to its enhanced dielectric breakdown strength and modulus of elasticity [196]. Recently, PVDF blends with various ILs have been established, offering tunability for applications ranging from biomedicine to energy [197]. These blends enhance the processability, induce specific crystalline phases, and fine-tune their optical and electrical properties. Table 7 lists the advantages and applications of PVDF blends with polymers.

Although PVDF composites exhibit multifunctional properties, many challenges must be addressed for practical applications, such as high dielectric loss and instability, strong temperature dependence, and functional separations from multifunction for desired device applications. These disadvantages might be due to (i) incorporations between organic PVDF and inorganic fillers, (ii) interface space charges between PVDF and fillers, etc. These disadvantages might not be fully reported in the literature, because positive phenomena are much easier to publish than are negative phenomena.

## 4. Material Property Characterizations and Techniques

Several characterization methods such as differential scanning calorimetry (DSC), Fourier transform infrared (FTIR) spectroscopy, and X-ray diffraction (XRD), have been widely used to measure the phase composition and understand the features of PVDF. Polarity switching measurements are then employed to determine the overall polarization of the material. Raman spectroscopy has recently been used for crystal structure. The mechanical properties can be measured through various tensile tests; the electrical properties of PVDF films can be characterized through electrical impedance spectra with impedance analyzers and others; the electromechanical properties can be achieved through a polarization electrical field loop (P-E loop), piezoelectric coefficient, electromechanical coupling factors, etc.; and the thermal electrical properties can be obtained by measuring the pyroelectric constants. This section focuses on the methods used to characterize the mechanical, electrical, thermal, structural, and electromechanical characteristics, as well as how they affect PVDF.

The characterization methods used to comprehend the fundamental properties of PVDF are shown in Figure 20 [204]. The complete crystallinity capability, which includes all non-amorphous phases that can be achieved by DSC, is shown in Figure 20a. Figure 20b illustrates the relative quantities of the crystalline phases determined using FTIR, XRD, and Raman spectroscopy. Figure 20c shows the P-E hysteresis for net polarization after the poling process. Table 8 presents popular techniques for characterizing PVDF, detailing its material, thermal, and electrical properties.

### 4.1. Thermal Characterization

Differential scanning calorimetry (DSC) and thermogravimetric analysis (TGA) are commonly used for thermal characterizations. DSC measures the heat flow associated with phase transitions such as melting and crystallization. It can be used to determine the crystallization temperature, melting temperature, and degree of crystallinity of the PVDF. In research, quality assurance, and industrial applications, DSC is frequently used for material inquiry, selection, comparison, and end-use evaluation of performance. The properties measured by TA Instruments’ DSC techniques include glass transitions, “cold” crystallization, phase changes, melting, crystallization, product stability, cure/cure kinetics, and oxidative stability. TGA measures the weight loss of a sample as a function of temperature. It can be used to determine the thermal stability of PVDF and identify potential degradation processes [205].

An example of PVDF engineering stress–strain curves under uniaxial tensile deformation at various temperatures is shown in Figure 21 [142]. The yield point is indicated by red arrows, the longest period’s greatest value is indicated by green arrows, the start of the α–β transition is indicated by blue arrows, and the moment at which the long period becomes stable and low after reaching its maximum is indicated by cyan arrows. The fact that the phase change started at almost the same strain at all temperatures is a striking phenomenon. The beginning of the α–β transition occurred at the PVDF’s orientational turning point. Figure 22 shows the temperature-dependent yield stress and strain [142]. The yield strains remain same, while the yield stress value decreases with the increase in temperature.

### 4.2. Structural Characterization

The two most used techniques for structural characterization are FTIR spectroscopy and XRD. XRD is used to determine the phase composition and crystal structure of the PVDF. This can help identify the presence of distinct crystalline phases, such as α, β, γ, and δ. FTIR spectroscopy is utilized to identify the functional groups present in the PVDF. It can also be employed to study the degree of crystallinity and the alignment of polymer chains [205,206,207].

An example of crystalline structural characterization using FTIR and wide-angle X-ray diffraction (WAXD) for biaxially oriented poly(vinylidene fluoride) (BOPVDF) is shown in Figure 23 [205]. The 2D WAXD patterns were used to create one-dimensional (1D) WAXD profiles at room temperature, as shown in Figure 23a. The machine direction (MD) is vertical, and the X-ray beam is in the transverse direction (TD). It also labels the main reflections for the α and β crystals where only the β crystal reflections (310)β, (110/200)β, (001)β, (220)β, (201)β, (221)β, and (311)β, were found for the strongly poled BOPVDF film. The FTIR spectra analysis for the poled and fresh BOPVDF films in transmission mode are displayed in Figure 23b, where the α and β crystal absorption bands are designated. According to this analysis, all the α crystals were transformed into β crystals by high-field electric poling, while the overall crystallinity remained unchanged. This was confirmed by FTIR measurements, which showed that all the α absorption bands vanished. Following lengthy unipolar poling at 650 MV/m, the BOPVDF film effectively produced pure β-phase crystals, according to the WAXD and FTIR data.

### 4.3. Mechanical Characterization

#### Tensile Testing

The tensile strength measures the resistance of PVDF to stress, while the Young’s modulus reflects its rigidity. Both are typically evaluated through tensile testing using a universal testing machine (UTM), where the sample thickness and internal stress conditioning are vital for accuracy.

Farusil et al. [208] used this apparatus for tensile and flexural testing, where the specimens were clamped using wedge grips with hooked jaws and subjected to uniaxial tensile loading at a rate of 1 mm/min. A custom-made guide provided precise longitudinal alignment of the clamped specimens with the vertical axis of the machine. Displacements were observed using a video extensometer. The load forces and displacements were measured until the specimens broke. The tensile strength was calculated by dividing the peak force by the cross-sectional area of the specimen. Figure 24 shows the tensile stress versus the tensile strain calculated for various pattern structures of the PVDF homopolymer and copolymer specimens.

To determine the compression modulus shown in Figure 25a in the normal direction of the BOPVDF sheet, a Nano Indent experiment was carried out using a Nano Indenter G200 (Agilent Technologies, Santa Clara, CA, USA) at a constant strain rate of 0.05 s^−1^ with a maximal indentation width of 1 μm [205]. The Young’s modulus of the BOPVDF film shown in Figure 25b was determined using tensile tests on a UTM (Model 5965, Instron Instruments, Norwood, MA, USA), in both the machining and the transverse directions. The highest values of k3j (j = 1, 2, and 3) are shown [205].

An example of the elastic modulus versus temperature graph for the PVDF film is shown in Figure 26. The elastic modulus is tested to calculate the elastic energy density, which is a crucial characteristic for many transducers. Furthermore, it displays the data for the stretched film exposed to 60 Mrad radiation at 95 °C along the drawing direction and the unstretched film exposed to 60 Mrad radiation at 120 °C. This shows that the elastic modulus of the stretched film Y along the drawing direction is significantly greater than that of the unstretched film. The stretched film’s modulus is 1.3 GPa at room temperature (approximately 20 °C), but the unstretched film’s modulus is 0.4 GPa [209].

### 4.4. Electrical Characterization

The dielectric constant measures the potential polarizations of PVDF to estimate the electrical energy generation and storage capabilities. LCR meters or LCR-based impedance analyzers are used to measure it across frequencies, with PVDF placed between the electrodes in a controlled electric field. Uniform electrode deposition (e.g., silver or aluminum) and controlled sample thickness are crucial for accuracy. Dielectric spectroscopy was used to study the dielectric properties, such as permittivity and dielectric loss, of PVDF. These properties are fundamental for understanding the ferroelectric and piezoelectric properties of PVDF. Figure 27 shows an example of the dielectric constant with respect to temperature in the range of −25 to 100 °C for PVDF-TrFE-CFE. The value of the dielectric constant increases with increasing temperature owing to the molecular motion and phase transition effect. As the temperature increases, the molecular motion enhances the dipole orientation owing to the applied electric field, which produces a higher dielectric constant for PVDF. PVDF features a variety of crystalline phases, including α, β, γ, and others. The phase transitions caused by temperature have the potential to drastically change the dielectric constant. Therefore, the dielectric constant significantly increases because of the transitions to the strongly polar β-phase. In addition, the impedance analyzer is a basic tool for piezoelectric device engineers. The impedance of a piezoelectric device should be monitored using an impedance analyzer throughout the device fabrication process.

#### Ferroelectricity

PVDF is a fascinating material owing to its unique ferroelectric properties. To understand these properties, we often visualize them using a polarization vs. electric field (P-E) loop. A P-E loop is a graphical representation of how a material’s polarization (electric dipole moment per unit volume) changes in response to an applied electric field. It is a bit like a hysteresis loop, showing how the material “remembers” its past state even after the external stimulus (electric field) is removed. The high-voltage dielectric and polarization loop test system is an example of a commercially available fully automated test system that enables consumers to measure the dielectric breakdown strength, consequently providing the charge–discharge efficiency and energy density and obtaining the polarization loop of ferroelectric and dielectric materials (www.piezopvdf.com). Another highlighted instrument utilized [205] for P-E loop measurement is the Premiere II ferroelectric tester (Radiant Technologies, Inc., Albuquerque, NM, USA).

Figure 28 shows that the P-E loop for commercial piezoelectric PVDF polymer, when subjected to an increasing electric field, exhibits a characteristic P-E loop (www.piezopvdf.com). The dipoles in the PVDF are randomly oriented as an electric field is applied. These dipoles start to align with the field, leading to an increase in polarization. At a certain field strength, most of the dipoles are aligned, and the polarization reaches a saturation point. When the electric field is reduced to zero, the dipoles do not completely return to their random orientations. A certain degree of polarization, known as remnant polarization (Pr), persists. To completely reverse the polarization, a reverse electric field called the coercive field (Ec) must be applied. The loop formed by polarization changes as the electric field is cycled known as the hysteresis loop. The area enclosed by this loop represents the energy loss during each cycle.

The P-E loop is a key parameter for measuring the ferroelectric properties of a ferroelectric material. Pr provides the piezoelectric properties of the piezoelectric material.

### 4.5. Electromechanical Properties

PVDF is a versatile polymer known for its unique electromechanical properties. Its capability to produce electrical energy from mechanical energy and vice versa makes it a significant material for various applications, including sensors, actuators, and energy-harvesting devices. Several techniques have been employed to characterize these properties.

#### 4.5.1. Piezoelectric Coefficient by Direct Methods

Researchers have utilized many techniques and equipment to characterize the piezoelectric coefficient of PVDF. Currently, the most popular procedures are the quasi-static, dynamic, interferometric, static, and acoustic testing methods.

The commercially available highlighted equipment includes “Berlincourt Quasi Static *d*_33_ m” (www.piezopvdf.com), and “Radiant’s Thin Film Piezoelectric Test Bundle” (www.ferrodevices.com). The characterization can be performed by either direct or converse piezoelectric effects methods.

##### Quasi-Static Method

The main setup of the quasi-static method for evaluating the piezoelectric properties of PVDF is a *d*_33_ m. The operational principle of this basic Berlincourt *d*_33_ m consists of multiple steps. First, the piezoelectric sample is mounted onto the probe of the meter and properly clamped. Second, a small oscillating force is applied to the sample through the probe. Finally, the electrical charge produced by the piezoelectric effect is then measured by the sensitive electronics meter, which is a charge-amplifier circuit. The *d*_33_ coefficient is calculated by dividing the measured charge by the applied force. This method involves gradually increasing the mechanical stress on the material and measuring the consequent electrical displacement. Several studies have focused on different measurement methods to investigate the influence of fabrication methods on the piezoelectric coefficient *d*_33_ of PVDF. For example, Satthiyaraju et al. [152] employed the quasi-static method to measure the *d*_33_ of annealed PVDF nanofibers, demonstrating that annealing significantly improved the piezoelectric performance by enhancing both the *d*_33_ coefficient and β-phase content.

The quasi-static method used Equations (1) and (2) to characterize the piezoelectric properties of the PVDF film [102].(1)$$Q=d_{33}A\sigma$$(2)$$d_{33}=Q_{sample}/F_{Dynamic}$$
where *Q* is the accumulated charge, $d_{33}$ is the piezoelectric coefficient, *A* is the sensing area, and σ is the applied stress.

In 1993, Hang et al. developed an equipment phase-sensitive *d*_33_ m and utilized the direct piezoelectric effect for the characterization of PVDF [210]. This high-sensitivity, phase-sensitive *d*_33_ m system was used to measure the *d*_33_ value of 1pC/N at phase angle 0.05°. However, its operational principle was based on the basic Berlincourt *d*_33_ m.

##### Dynamic Method

Unlike the quasi-static approach, the dynamic method applies a high-frequency force to the material to measure the resulting voltage. The setup involves placing the sample on a mass connected to an exciter that provides sinusoidal acceleration. This configuration evaluates the piezoelectric coefficient by varying parameters such as the frequency and amplitude, making it suitable for sensor and actuator applications. Dynamic methods are divided into cyclic loading and vibration modal techniques that utilize the transient effect of PVDF to generate regular electrical signals [211]. The dynamic method is governed by the relationship between the applied stress (*σ*), the generated electric field (*E*), and the frequency of the applied force (f). The piezoelectric charge constant (*d_ij_*) and voltage response (*g_ij_*) can be expressed by Equations (3) and (4):(3)$$d_{ij}=\frac{Q}{\sigma }$$(4)$$g_{ij}=\frac{E}{\sigma }$$
where *Q* is the charge generated by the material, *σ* is the applied oscillatory stress, *E* is the electric field generated in response to the applied stress, $d_{ij}$ is the piezoelectric charge constant, and $g_{ij}$ represents the voltage response.

In the dynamic method, the total pressure *p*(*t*), shown in Equation (5), on the PVDF sample is the sum of the static load $P_{0}$ and the dynamic oscillating load ${\hat{p}}_{d}\cos\left(\omega t\right)$, where *m* is the mass, *A* is the sample area, *g* is gravity, and $g+\hat{a}\cos\left(\omega t\right)$ represents the sinusoidal acceleration. The piezoelectric coefficient *d*_33_ is calculated by measuring the voltage response to these forces, which vary with frequency and amplitude. Although dynamic tests typically yield a slightly lower *d*_33_ value, they provide valuable insights into the sensor and actuator performance of PVDF.(5)$$p\left(t\right)=\frac{m}{A}\left(g+\hat{a}\cos\left(\omega t\right)\right)=p_{0}+{\hat{p}}_{d}\cos\left(\omega t\right)$$

Mrllík et al. [55] used dynamic testing to investigate the piezoelectric properties of PVDF films, focusing on the influence of harmonic oscillations on the electric charge response of the material, ultimately demonstrating improved sensitivity in vibration-sensing applications. Aghayari et al. [212] used dynamic testing to assess piezoelectric properties of PVDF under cyclic loading, revealing enhanced electrical performance and stability, making the material suitable for applications in sensor technologies.

##### Acoustic Method

In the acoustic method, the piezoelectric material acts as a microphone, and its sensitivity is used to determine the piezoelectric coefficient. This method involves placing the sample on a back electrode and subjecting it to sound waves in an anechoic chamber to avoid external noise. The sensitivity of the microphone is calculated, which, together with the capacitance and area of the sample, allows for the determination of the *d*_33_ coefficient. Although effective, the need for an anechoic chamber and the associated costs make this method more complex and expensive.

The primary equation used in the acoustic method is based on the relationship between the sound pressure (*P*), piezoelectric charge constant (*d*), resulting electric field (*E*), and induced strain (*S*) by the applied pressure as given in Equation (6).(6)P=Ed×S

The acoustic method is particularly useful for measuring the piezoelectric charge constant (*d*) in applications involving acoustic stimuli. This method can also be adapted to measure the voltage response of PVDF, which is crucial for understanding the performance of materials in acoustic sensor applications. The electric displacement, which represents the charge per unit area generated by the material in response to the sound pressure, can also be measured using the acoustic method. This measurement is critical for the design of PVDF-based acoustic sensors.

Figure 29 shows an example of the piezoelectric characteristics of PVDF, in which several piezoelectric coefficients are calculated by direct piezoelectric testing. Figure 29a shows that *d*_33_ is negative, and its absolute value increases as the applied dynamic stress increases. It is evident that a typical |*d*_33_| of ~18 pC/N is attained below 0.1 MPa. The *d*_33_ value obtained using a *d*_33_ piezometer with a static load of 2.5 N is indicated by a red star. Figure 29b shows the stress-dependent measurements of *d*_31_ and *d*_32_. At 41 MPa, the highest values are *d*_31_ = 22 pC/N, and at 49 MPa, *d*_32_ = 18 pC/N [205].

#### 4.5.2. Strain Induced by Indirect Piezoelectric Effect

##### Interferometric Method

The interferometric method utilizes laser interferometry based on the inverse piezoelectric effect. Although this method is less commonly used than quasi-static and dynamic methods, it offers a high resolution and is more effective over a broad frequency range [213]. By measuring the vibration amplitude of the sample, the coefficient is derived from the ratio of this amplitude to the applied voltage, which can be calculated using Equation (7):(7)$$d_{33}=\frac{∆L}{\Delta F}$$
where Δ*L* is the change in length of the sample, *L* is the original length, and Δ*F* is the applied force.

The interferometric method is particularly effective for accurately measuring the piezoelectric charge constant (*d_ij_*), particularly in applications that require high precision. Chen et al. [214] used interferometric techniques to investigate the electromechanical behavior of PVDF-TrFE-FA terpolymers, revealing a significant improvement in the electroactuation strain at low electric fields, achieving up to −3.3% strain at 50 MV/m. Chen et al. [215] utilized interferometric techniques to investigate the electromechanical properties of PVDF-based polymers, achieving an impressive *d*_33_ of −1050 pm/V and a coupling factor k_33_ of 88%, demonstrating high electroactuation at low fields. An example of the PVDF strain measured using a dilatometer based on a cantilever beam can be found in the reference [216]. The dilatometer is simple to use and can measure the transverse strain response of soft polymer films without requiring the sample to be mechanically constrained over a wide strain range. Under various load and temperature conditions, the strain can be measured across a broad frequency range, from mHz to over 100 Hz. The transverse strain of a pure PVDF piezoelectric film at room temperature under various electric fields of 1 Hz is shown in Figure 30. The frequency dependence of the transverse strain of the PVDF piezoelectric film caused by the electric field is shown in Figure 31. Figure 31 shows the load impact on the electric field that caused the transverse strain response for a 65/35 mol% PVDF-TrFE copolymer film that was exposed to radiation at 95 °C with a dose of 60 Mrad at room temperature with an applied electric field of 1 Hz. The relationship between the electric field and strain response for the films under various loads is shown in Figure 32a. The relationship between the strain response and static load for the films at varying electric field strengths is depicted in Figure 32b.

#### 4.5.3. Resonance Methods

The resonance method is not commonly used by in the material scientist community. However, it is a convenient method for ferroelectric device engineers to evaluate piezoelectric PVDF films, which can be used for device fabrication. For a piezoelectric-state PVDF film, for instance, a stretched and poled PVDF film, an impedance analyzer can conduct the measurement. However, for the PVDF film without stretching and poling, a DC bias field must be applied to convert the PVDF film to the piezoelectric state [217].

The coupling factor of a piezoelectric material can also be obtained using the resonance technique, which involves measuring the electrical impedance as a function of frequency near a specified resonance frequency [218]. The frequency variation of the dielectric properties of the stretched and irradiated P(VDF-TrFE) copolymer is illustrated in Figure 33 [217]. The statistics indicate that both the dielectric constant and the loss diminish with the application of a DC bias field. A distinct resonance is observed in the frequency range of 20–50 kHz, corresponding to the resonance along the stretching direction of the sample. The resonance intensifies with an increase in the DC bias field. The impedance and admittance of the sample can be computed based on the dielectric behavior show in Figure 33. The series resonance frequency *f_s_*, corresponding to the maximal real component (*R*) of the impedance, and the parallel resonance frequency *f_p_*, corresponding to the maximal real component (*G*) of the admittance, can be determined from the frequency-dependent values of *R* and *G*. According to the standard procedure (IEEE Standard, 1988 [219]),(8)$$\frac{k_{31}^{2}}{1-k_{31}^{2}}=\frac{\pi }{2}\frac{f_{p}}{f_{s}}\tan\frac{\pi }{2}\frac{f_{p}-f_{s}}{f_{s}}$$(9)$$\frac{1}{S_{11}^{E}}=4\rho f_{s}l$$
where $S_{11}^{E}$ represents the elastic compliance in the stretching direction, ρ is the density, and l is the length in the resonance direction. The electromechanical coupling coefficient, *k*_31_, of the specimen under various DC bias fields is computed, as illustrated in Figure 34 [217]. The results demonstrate that the coupling factor in the sample remains constant across frequencies.

### 4.6. Pyroelectric Coefficient

The pyroelectric coefficient represents the electric charge generated by temperature fluctuations, which is crucial for thermal and infrared sensors. The measurement methods for the pyroelectric coefficient include direct measurement using a heating source and an electrometer to record charge changes, with smooth electrodes ensuring uniform heating. Dynamic measurement can also capture the pyroelectric current response to periodic temperature changes. The pyroelectric coefficient (*p*) was calculated as *p* = Δ*QA*·Δ*T*, where Δ*Q* is the charge generated, A is the electrode area, and Δ*T* is the temperature change. The pyroelectric current (*I*) is given by *I* = *p*·*A·*d*T*d*t*, where d*T*/d*t* is the rate of temperature change [122,220]. Examples of pyroelectric PVDF films are shown in Figure 35 and Figure 36 [121].

Figure 35 illustrates the maximum pyroelectric coefficient achieved with 1 wt% doping of graphene-oxide (GO). GO-doping significantly enhances the pyroelectric characteristics of PVDF. Figure 36 shows that the pyroelectric current for the 1 wt% GO-doping scenario is three times that of the 2 wt% GO-doping scenario and tenfold that of the scenario without GO-doping. Excessive GO-doping results in current leakage. High doping concentrations have been shown to reduce the figure of merit (FV) due to a substantial increase in the dielectric constant, resulting in a diminished pyroelectric signal.

## 5. Summary and Future Perspectives

This paper provides a comprehensive review of PVDF polymer films and their material properties, fabrication methodologies, and characterization methods. The physical, mechanical, chemical, thermal, electrical, and electromechanical properties of PVDF are described in detail. Special focus is given to PVDF film fabrication methods, which include solution casting and coating, spin coating, hot pressing, electrospinning, and 3D printing. Techniques such as mechanical stretching, annealing, and electric field poling can enhance phase transition, resulting in improved piezoelectric and pyroelectric properties. Moreover, this article explores the characterization of PVDF material for its thermal, structural, mechanical, electrical, and electromechanical properties, using various methods such as FTIR, XRD, DSC, PFM, UTM, TGA, dielectric test systems, ferroelectric test systems, piezoelectric test methods, dynamic measurement methods, electric field induced strain and actuation measurement methods, electromechanical coupling factor measurements, and pyroelectric coefficient measurements, providing insights into its properties and performance in various applications. Understanding these aspects can help optimize PVDF-based devices for energy harvesting, sensing, biomedical applications, radio-electronic devices, architectural coating, printed circuit boards, non-destructive testing, environmental monitoring, and many more, driving advancements in these fields.

Films are the most popular shapes used for PVDF materials. Solution casting and coating are the most popular methods for fabricating PVDF films in lab-scale studies. Spin coating provides more uniform PVDF films for micro-electromechanical system applications integrated with silicon wafer-based electronics. The hot press method provides a relatively high-quality piezoelectric film for large-scale commercial fabrication. However, the hot press method makes it difficult to fabricate unformed films in small lab-scale fabrication. For pure PVDF films, the post-film stretching process to transfer the α-phase to the β-phase, annealing to increase crystallinity, and electrical poling to form permanent polarization are needed to improve the ferroelectric and piezoelectric properties. In film processing, (i) DSC is the basic method to check the melting, glass transition, and crystallization temperatures to determine the annealing conditions, while the thermal stability can be measured with TGA; (ii) WAXD, XRD, and FTIR are used to determine the β-phase; (iii) an impedance analyzer for dielectric measurements and a P-E loop for polarization measurement are used to determine the electrical, dielectric, and ferroelectric properties of the film; (iv) the mechanical properties can be measured with UTM and DMA; (v) the direct piezoelectric effect can be determined with *d*_33_ m, PFM, quasi-static and dynamic, and acoustic excitation measurements; (vi) the inverse piezoelectric effects can be measured by various electrical field-induced strain measurement methods; and (vii) the electromechanical properties can be determined by the ratio of output mechanical energy over input electrical energy or output electrical energy over the input mechanical energy. In addition, the major electromechanical properties can be measured using the impedance resonance method.

Currently, piezoelectric PVDF films are still dominant in practical applications because they are well studied and mature. Although various new PVDF-based copolymers, terpolymers, and nanocomposites have been developed recently, most of these materials have not been comprehensively characterized for device applications. Collaborations among material scientists, mechanical and electrical engineering, and device developers may accelerate research in this field. The adoption of ferroelectric ceramic research methods in PVDF research is another approach.

Many PVDF composites exhibit multifunctional properties; however, many challenges have not been addressed for practical applications, such as high dielectric loss and instability, strong temperature dependence, and functional separation from multiple functions for desired device applications. Many disadvantages might not be fully reported in the literature because positive phenomena are much easier to publish than are negative phenomena. More detailed studies are needed in the near future.

Considering the length of this paper, we do not include a review of PVDF-based devices, such as sensors, actuators, transducers, and energy harvesters. We will provide a comprehensive review of specific device issues in the near future.

## Figures and Tables

**Figure 1 micromachines-16-00386-f001:**
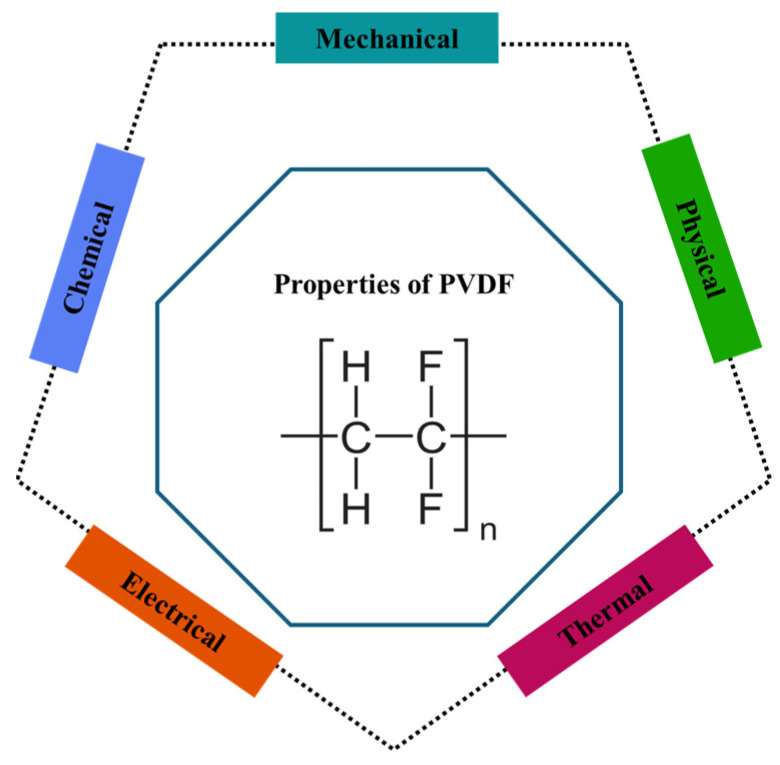
Different properties of PVDF.

**Figure 2 micromachines-16-00386-f002:**
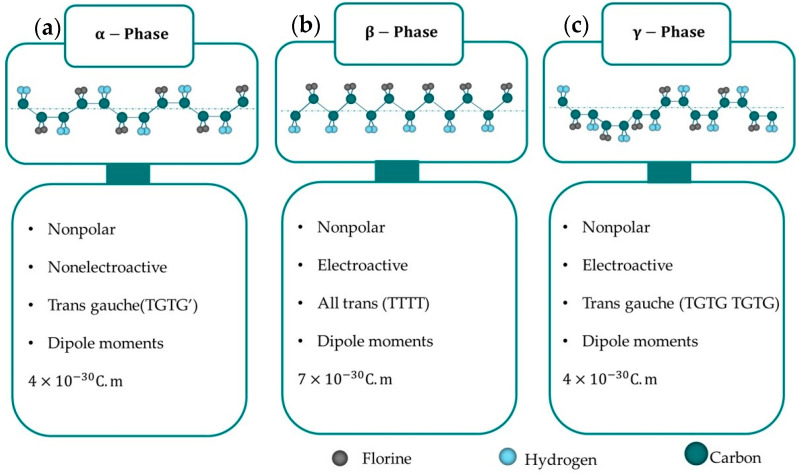
Different phases of PVDF and their properties. (**a**) α-phase, (**b**) β-phase, and (**c**) γ-phase. Redrawn from reference [11,74].

**Figure 3 micromachines-16-00386-f003:**
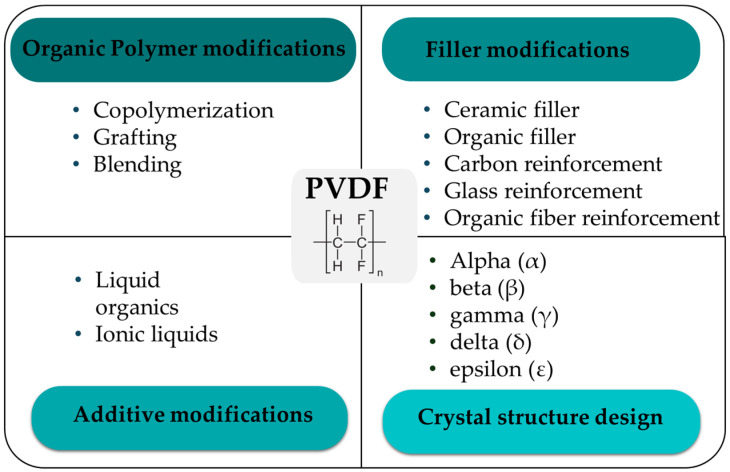
Different methods of improving PVDF properties.

**Figure 4 micromachines-16-00386-f004:**
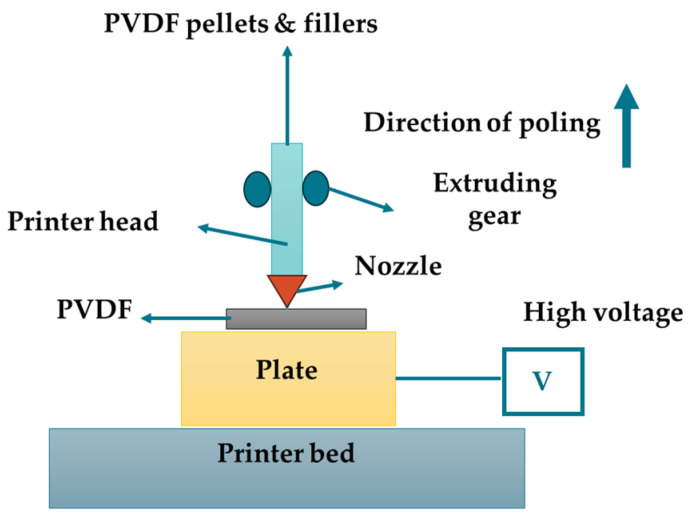
A 3D printing mechanism for PVDF films.

**Figure 5 micromachines-16-00386-f005:**
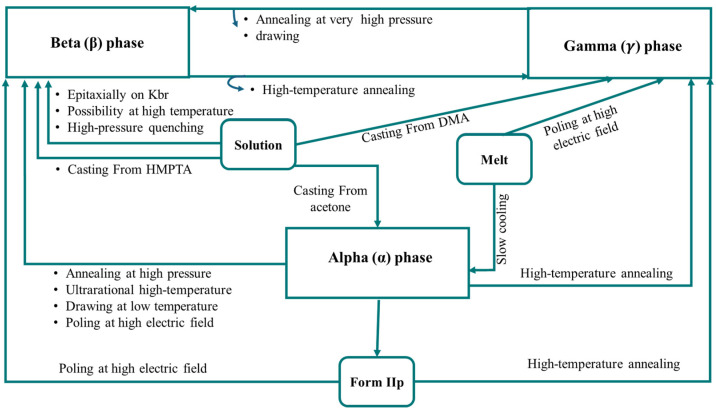
The PVDF polymer’s crystalline phase transformation diagram [140].

**Figure 6 micromachines-16-00386-f006:**
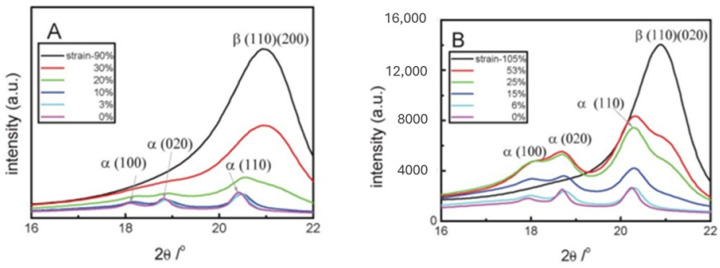
Phase transition by temperature stretching method. The 1D WAXS profiles during stretching at (**A**) 60 °C and (**B**) 140 °C [142].

**Figure 7 micromachines-16-00386-f007:**
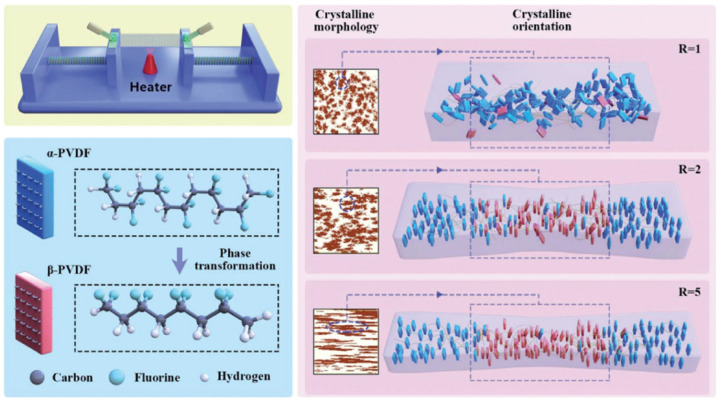
Diagrammatic representation of PVDF films’ uniaxial stretching process and structure formation phase change, crystalline orientation, and crystalline shape as the stretch ratio increases [145].

**Figure 8 micromachines-16-00386-f008:**
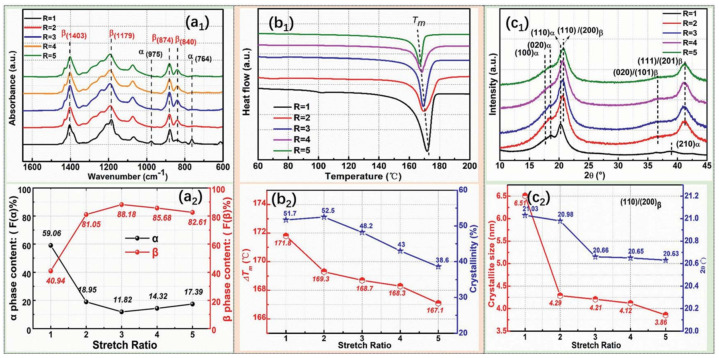
Various stretching conditions of PVDF films for structure evolution. (**a_1_**) FTIR spectra. (**a_2_**) PVDF films’ phase composition changes when the stretch ratio rises. (**b_1_**) DSC traces. (**b_2_**) Crystallinity and melting temperatures in relation to stretch ratios. (**c_1_**) The 1D WAXD patterns. (**c_2_**) The (110)/(200)β-phase’s crystallite size and 2θ as a function of stretch ratio, determined by 1D XRD [145].

**Figure 9 micromachines-16-00386-f009:**
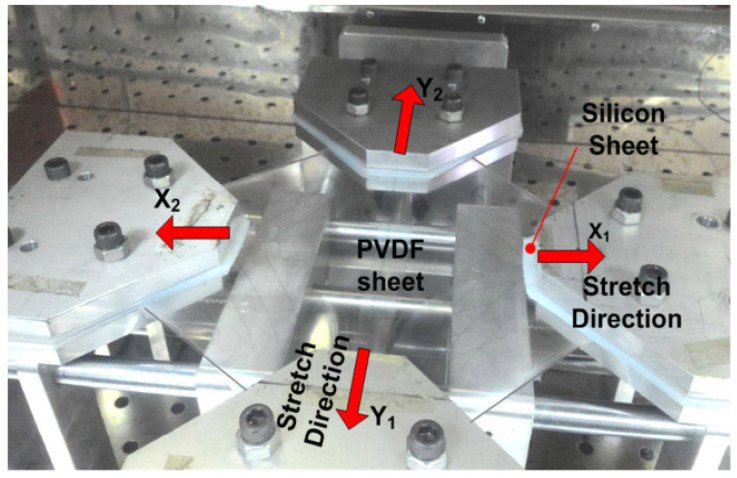
Experimental setup for biaxial stretching of PVDF [104].

**Figure 10 micromachines-16-00386-f010:**
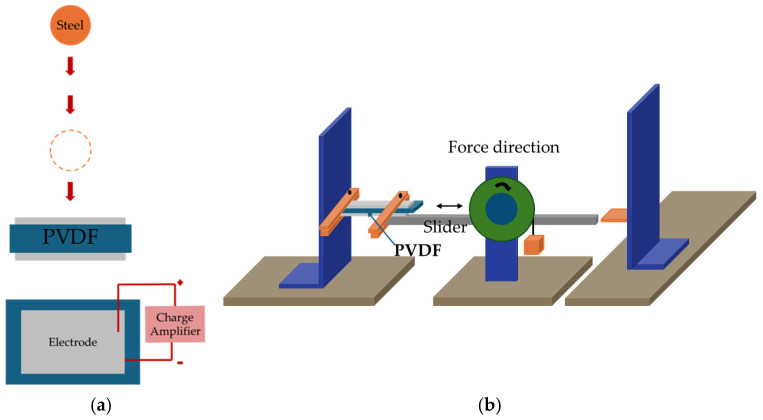
Test setup for measurement of piezoelectric coefficient: (**a**) *d*_33_ and (**b**) *d*_31_ [104].

**Figure 11 micromachines-16-00386-f011:**
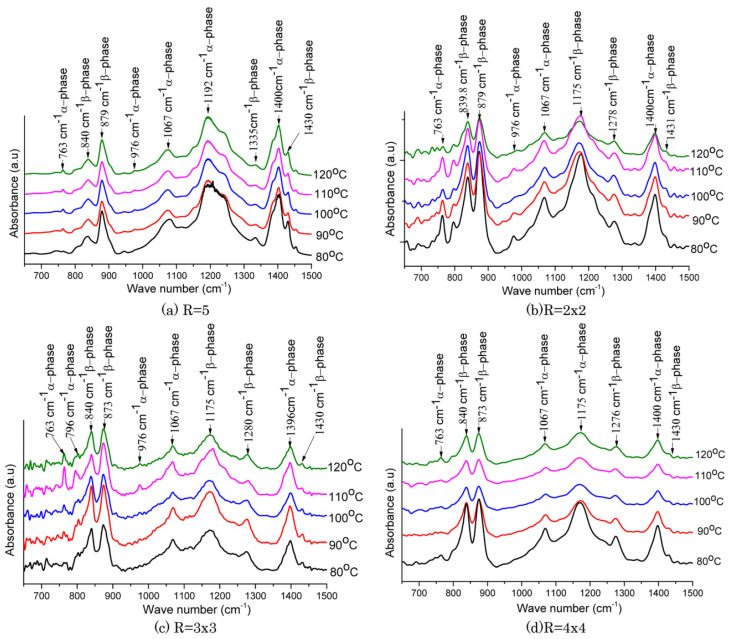
FTIR spectra of stretched PVDF: (**a**) uniaxial and (**b**–**d**) biaxial [104].

**Figure 12 micromachines-16-00386-f012:**
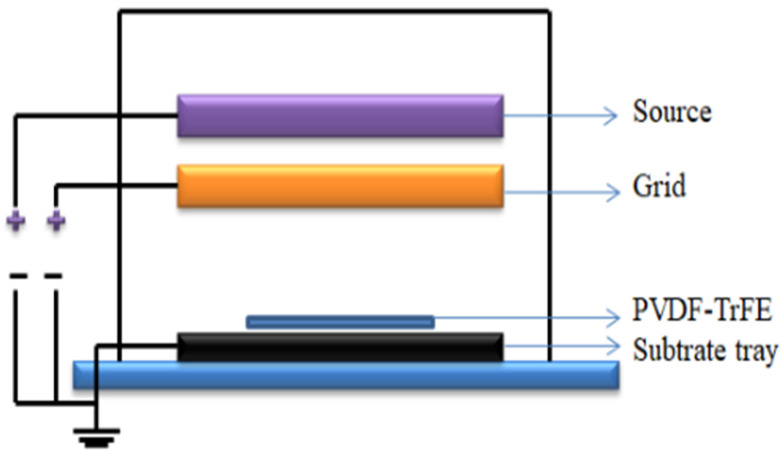
Schematic diagram of in situ polarization for PVDF film [156].

**Figure 13 micromachines-16-00386-f013:**
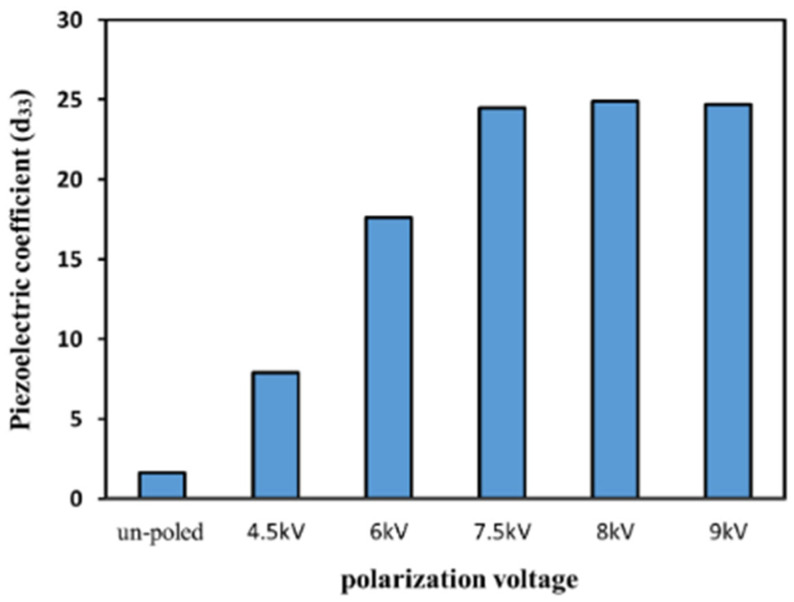
Piezoelectric coefficient corresponding to different polarization voltage [156].

**Figure 14 micromachines-16-00386-f014:**
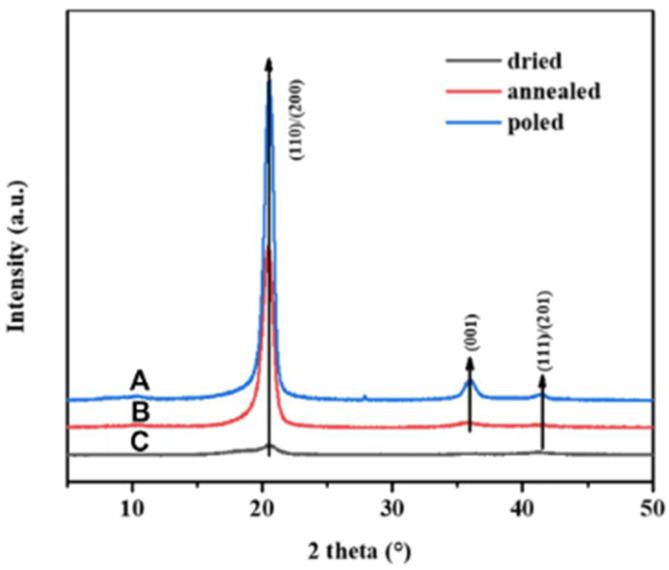
XRD patterns of PVDF-TrFE films after poling [9].

**Figure 15 micromachines-16-00386-f015:**
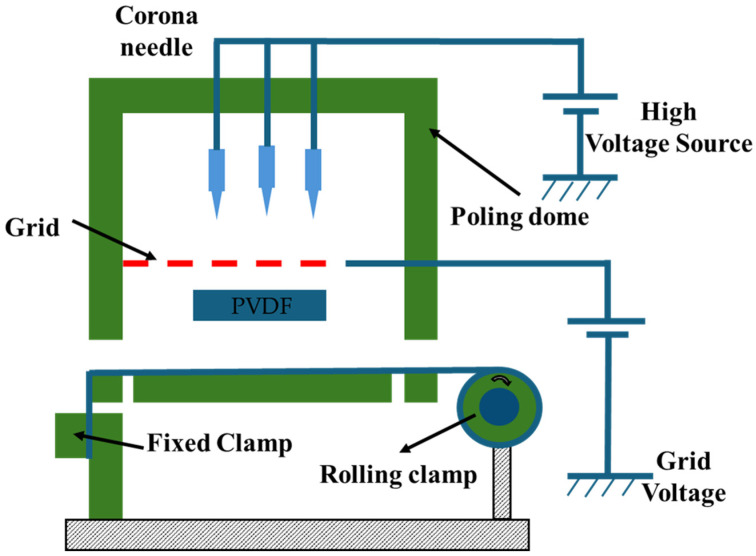
Schematic of setup for stretching and corona poling experiment [157].

**Figure 16 micromachines-16-00386-f016:**
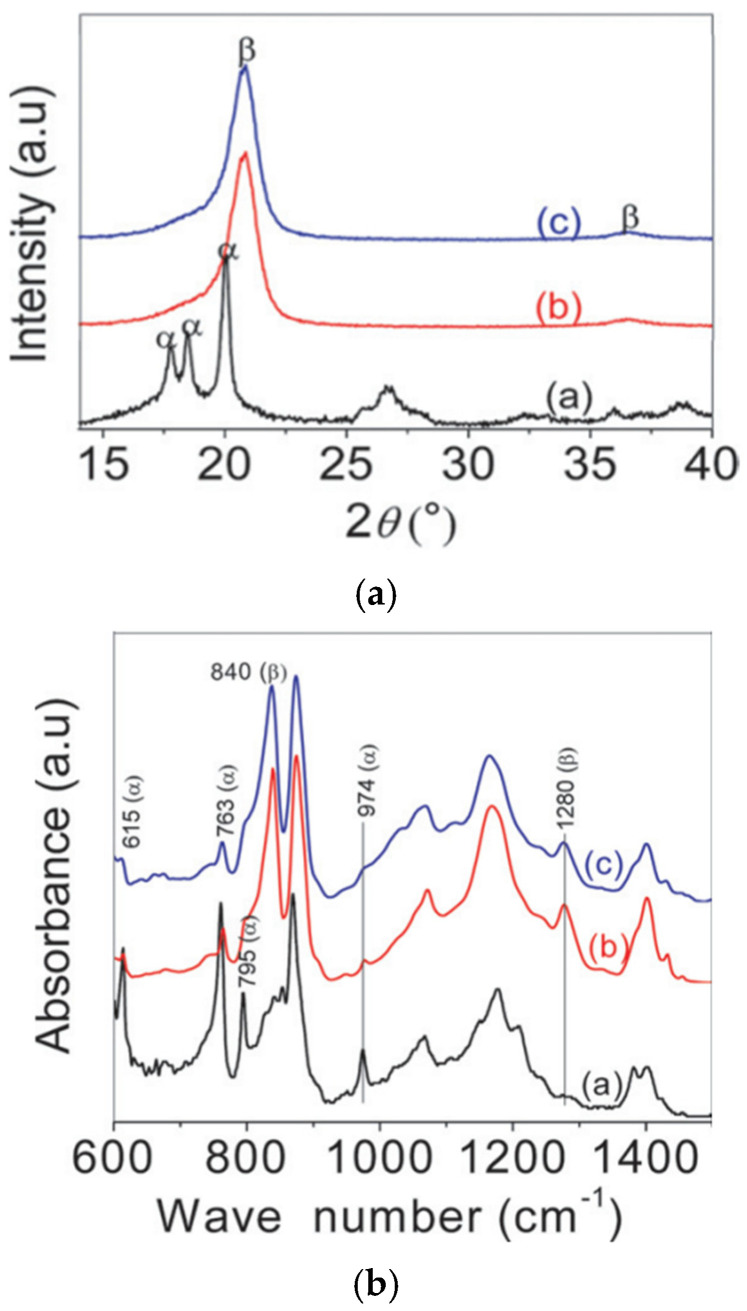
(**a**) XRD patterns of PVDF after corona polarization. (**b**) FTIR spectra of PVDF [157].

**Figure 17 micromachines-16-00386-f017:**
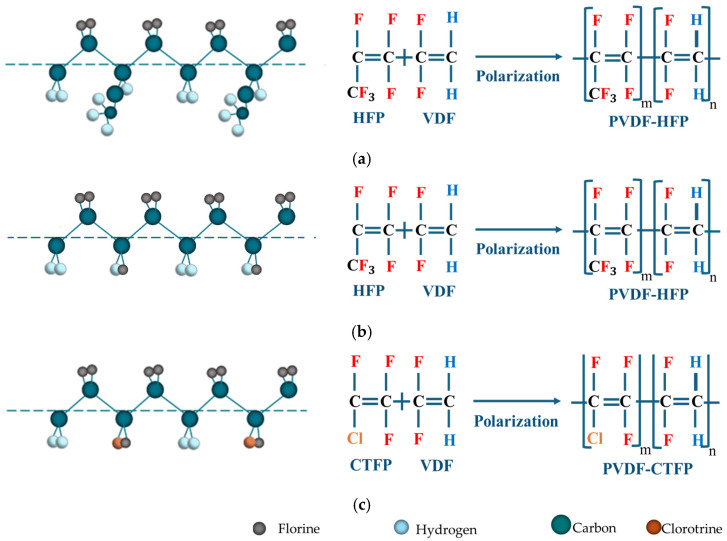
PVDF copolymers: (**a**) poly(vinylidene fluoride-hexafluoropropylene), (**b**) poly(vinylidene fluoride-co-trifluoroethylene), and (**c**) poly(vinylidene fluoride-co-chlorotrifluoroethylene).

**Figure 18 micromachines-16-00386-f018:**
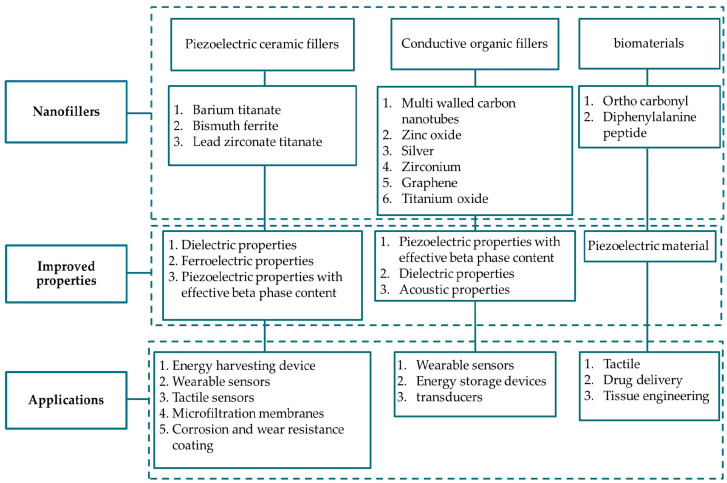
Effects of various nanofillers on PVDF properties and their applications.

**Figure 19 micromachines-16-00386-f019:**
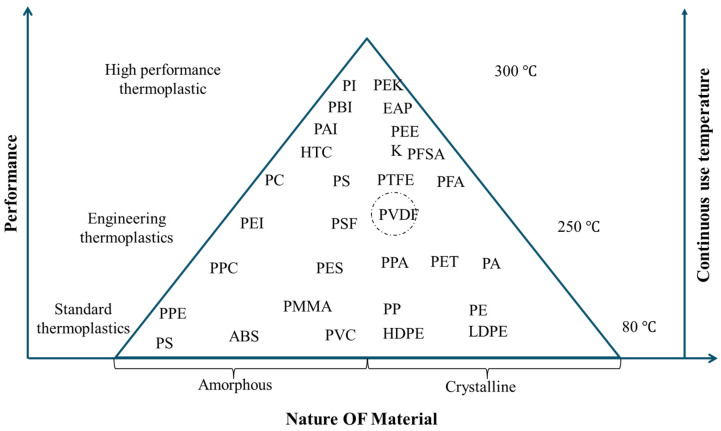
The classification of thermoplastics blending criteria [189].

**Figure 20 micromachines-16-00386-f020:**
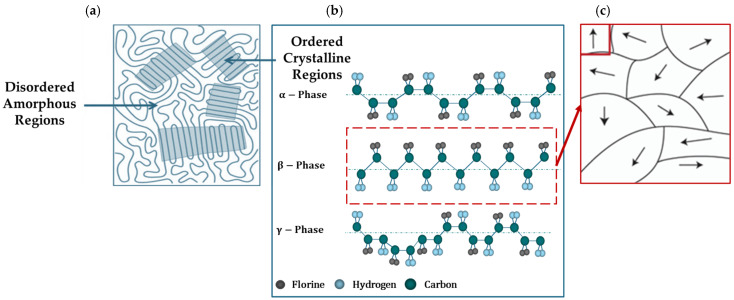
Different methods used for PVDF characterization. (**a**) Crystalline and amorphous regions measured with DSC. (**b**) Crystallin phases, measured with FTIR, XRD, and Raman. (**c**) Net polarization P-E hysteresis measured with a ferroelectric test station [204].

**Figure 21 micromachines-16-00386-f021:**
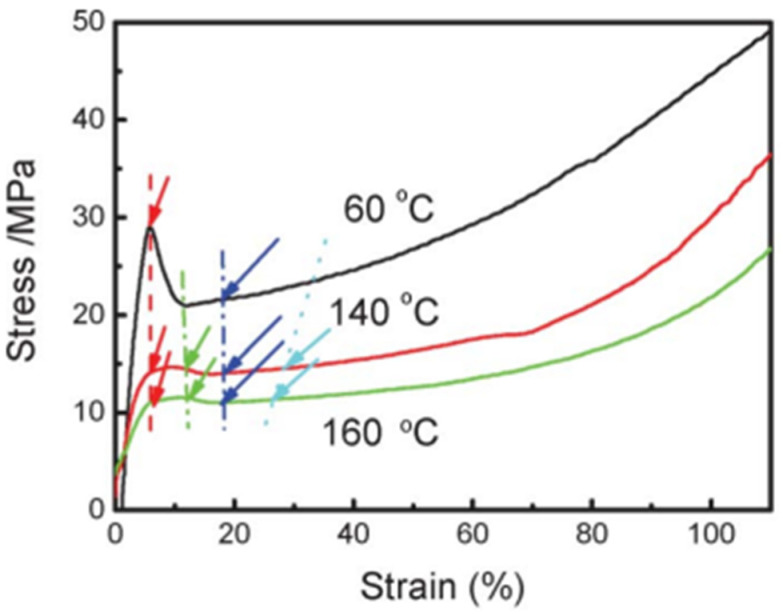
PVDF’s engineering stress–strain curves at various temperatures under uniaxial tensile deformation [142].

**Figure 22 micromachines-16-00386-f022:**
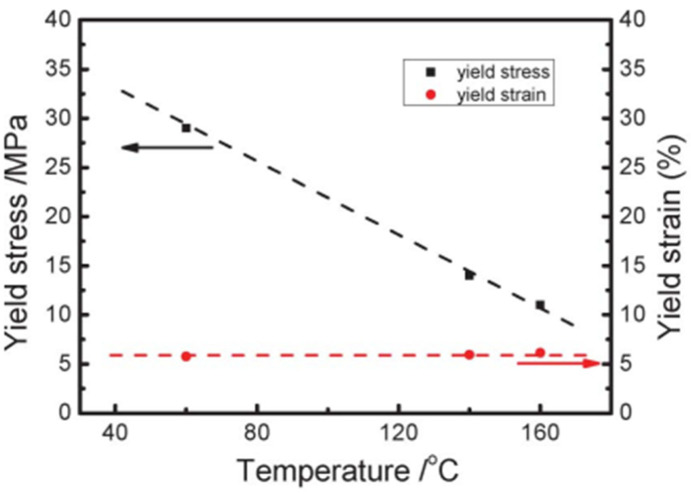
Yield stress and yield strain at various temperatures [142].

**Figure 23 micromachines-16-00386-f023:**
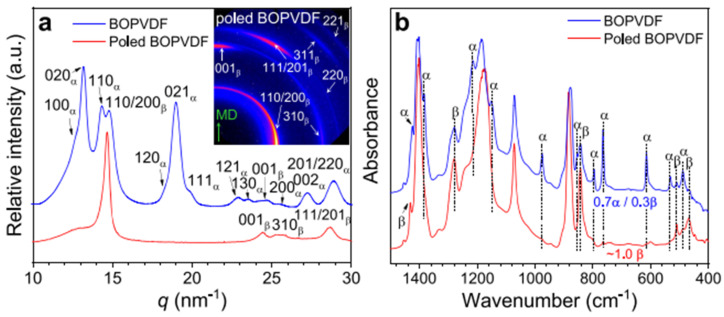
Structural characterizations by WAXD and FTIR. (**a**) BOPVDF films at room temperature are profiled using 1D WAXD. (**b**) The FTIR spectra analysis for the poled and fresh BOPVDF films in the transmission mode [205].

**Figure 24 micromachines-16-00386-f024:**
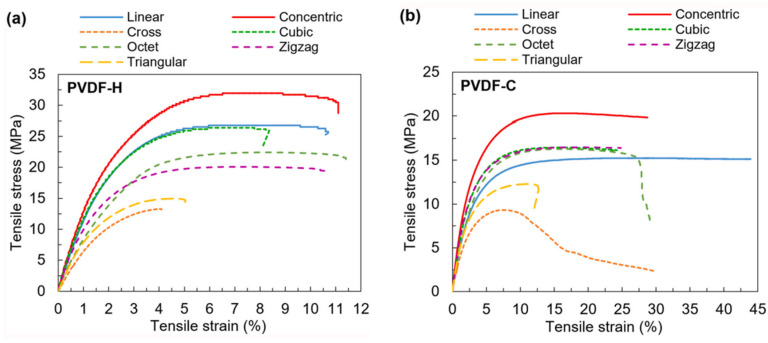
Tensile stress versus tensile strain curves of specimens printed with various crystalline patterns by using (**a**) PVDF-homopolymer and (**b**) PVDF-copolymer [208].

**Figure 25 micromachines-16-00386-f025:**
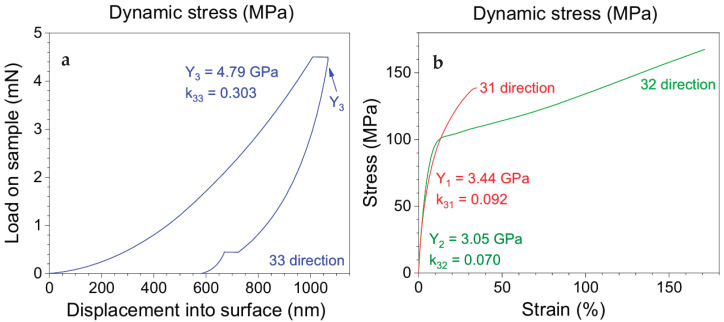
Tensile test for highly poled PVDF film. (**a**) The compression modulus (Y3) is obtained by nanoindentation. (**b**) Tensile moduli (Y1 and Y2) are obtained from the stress–strain curves [205].

**Figure 26 micromachines-16-00386-f026:**
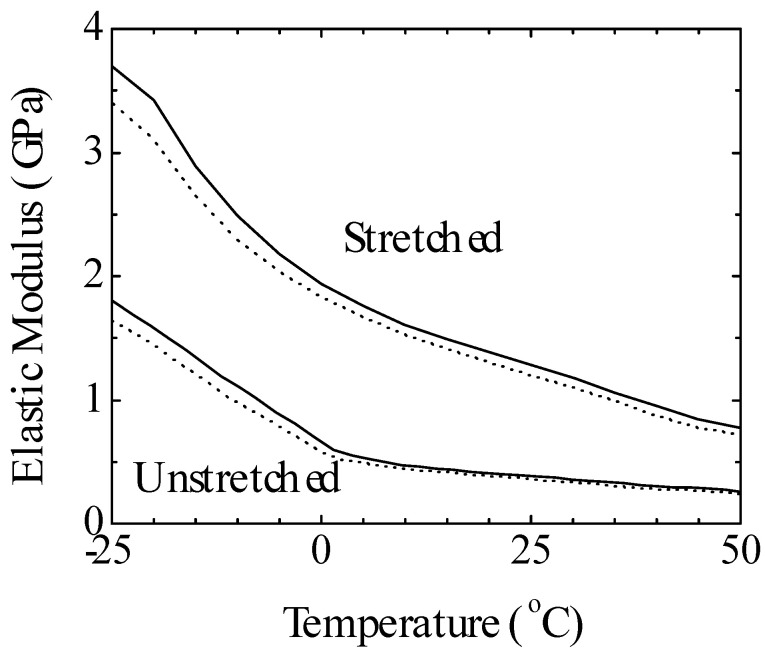
The elastic modulus for both stretched and unstretched materials as a function of temperature. The data recorded at 1 and 10 Hz are represented by the dots and solid lines, respectively [209].

**Figure 27 micromachines-16-00386-f027:**
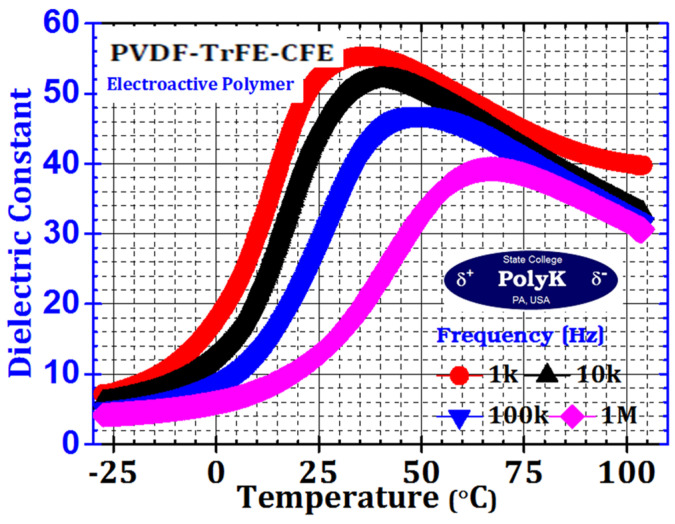
Dielectric constant for various temperature of PVDF-TrFE-CFE [57].

**Figure 28 micromachines-16-00386-f028:**
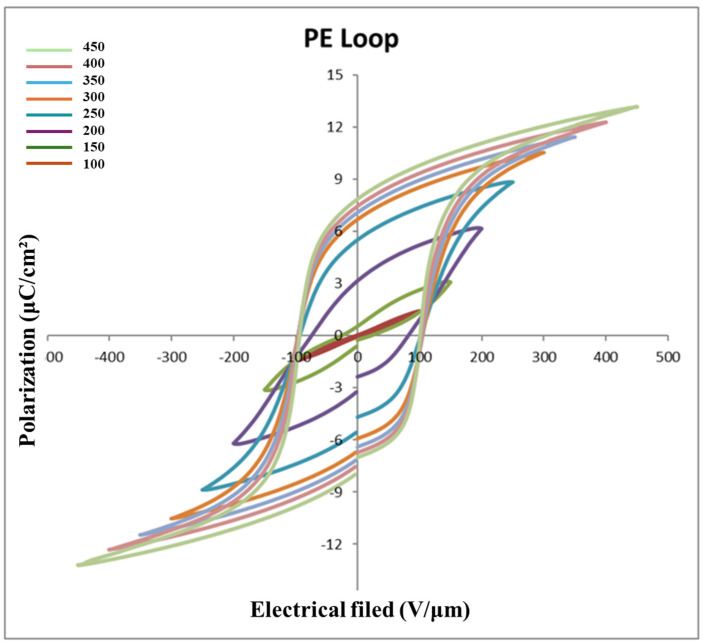
P-E loop for commercial piezoelectric PVDF polymer provided by PolyK Technologies [57].

**Figure 29 micromachines-16-00386-f029:**
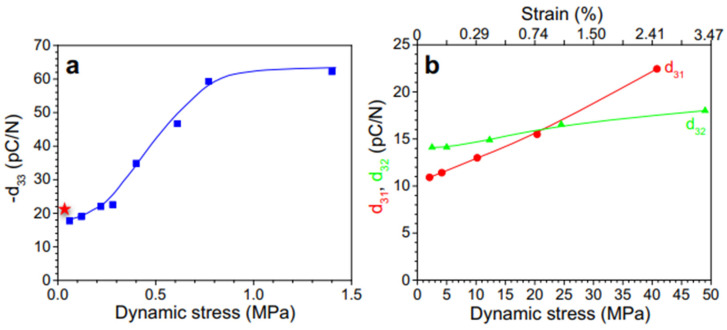
Several piezoelectric coefficients for highly poled BOPVDF film: (**a**) *d*_33_ and (**b**) *d*_31_ and *d*_32_ as a function of dynamic stress [205]. The red star in (**a**) indicates the *d*_33_ value measured by the *d*_33_ piezo meter with a static force of 2.5 N.

**Figure 30 micromachines-16-00386-f030:**
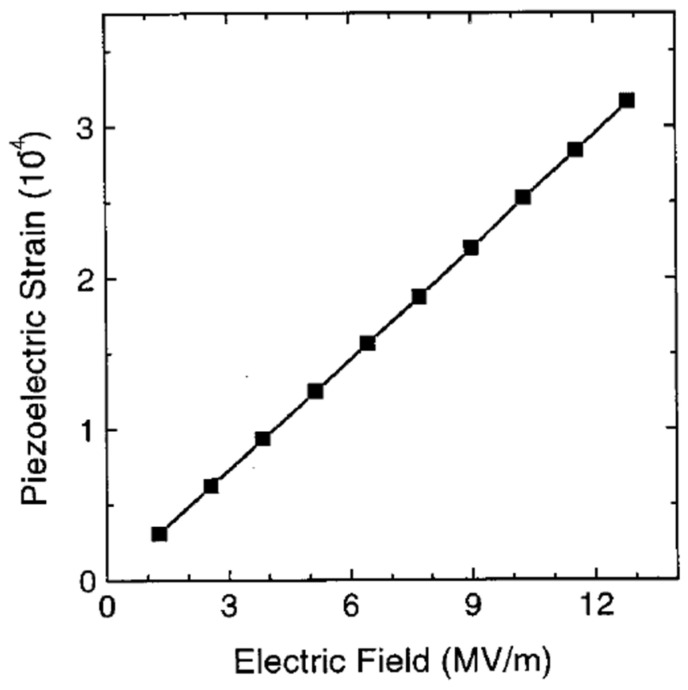
Pure PVDF piezoelectric film’s transverse strain at room temperature under various electric fields of 1 Hz [216].

**Figure 31 micromachines-16-00386-f031:**
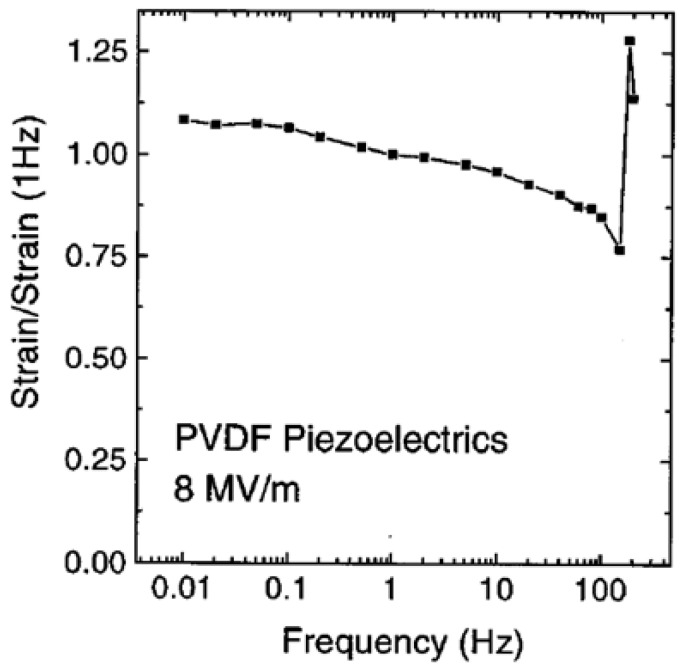
Frequency dependence of the transverse strain caused by an electric field in a PVDF piezoelectric film [216].

**Figure 32 micromachines-16-00386-f032:**
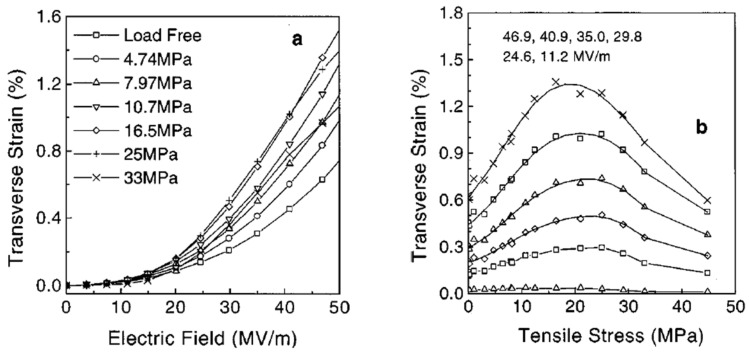
Load effect on the electric field induced transverse strain response measured for PVDF-TrFE copolymer film. (**a**) Strain response as function of electric field for films under different loads. (**b**) Strain response and static load for films at various electric field strengths [216].

**Figure 33 micromachines-16-00386-f033:**
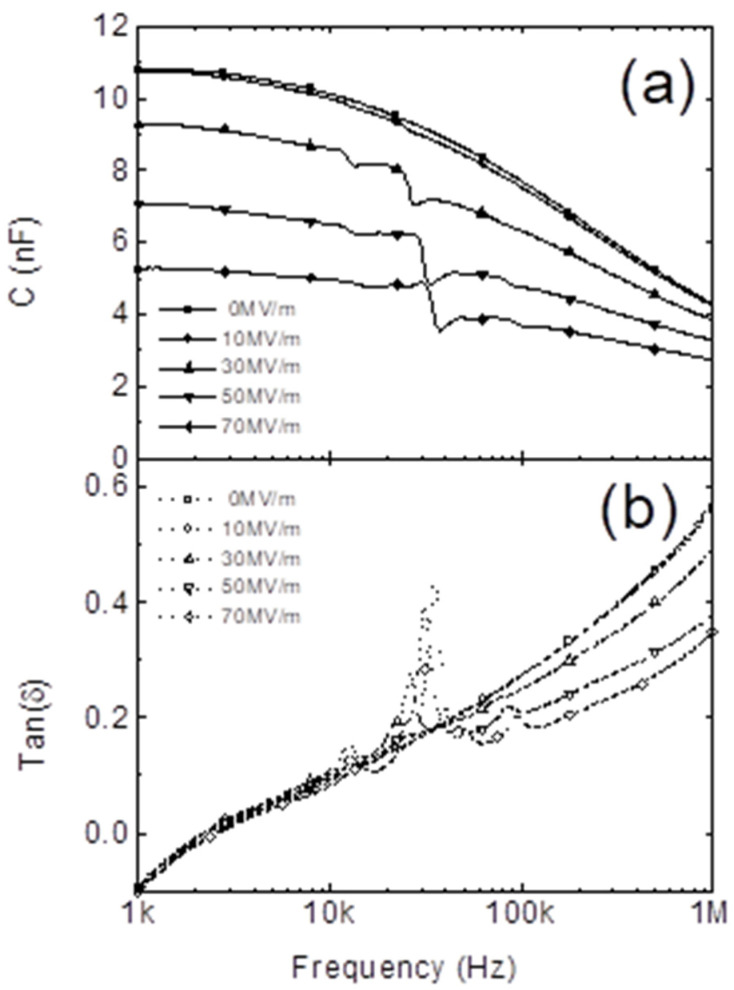
The impedance as a function of frequency: (**a**) the capacitance and (**b**) the dielectric loss [217].

**Figure 34 micromachines-16-00386-f034:**
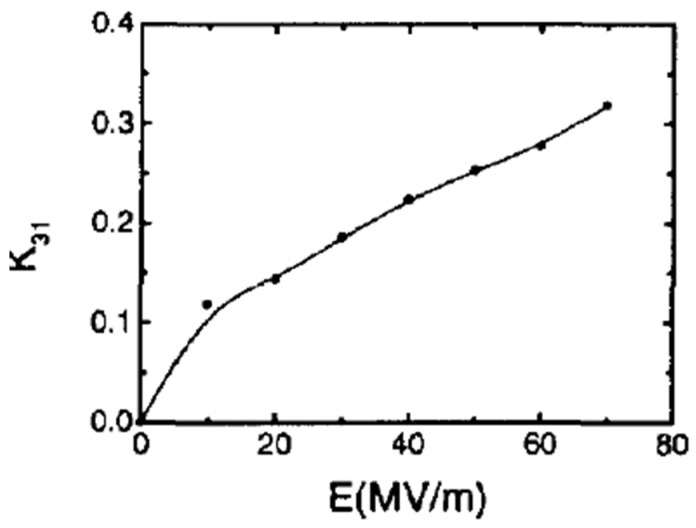
Electromechanical coupling coefficient vs. dc-bias fields for the copolymer [217].

**Figure 35 micromachines-16-00386-f035:**
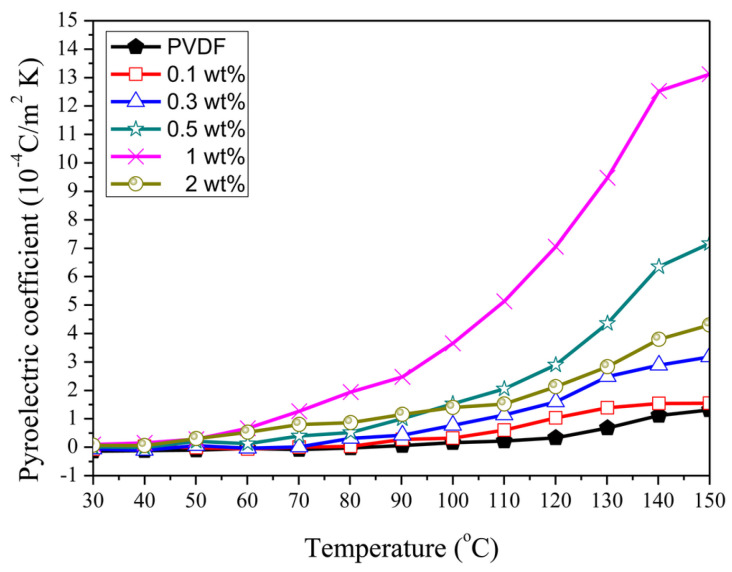
The pyroelectricities of PVDF film with various wt% GO-doping [122].

**Figure 36 micromachines-16-00386-f036:**
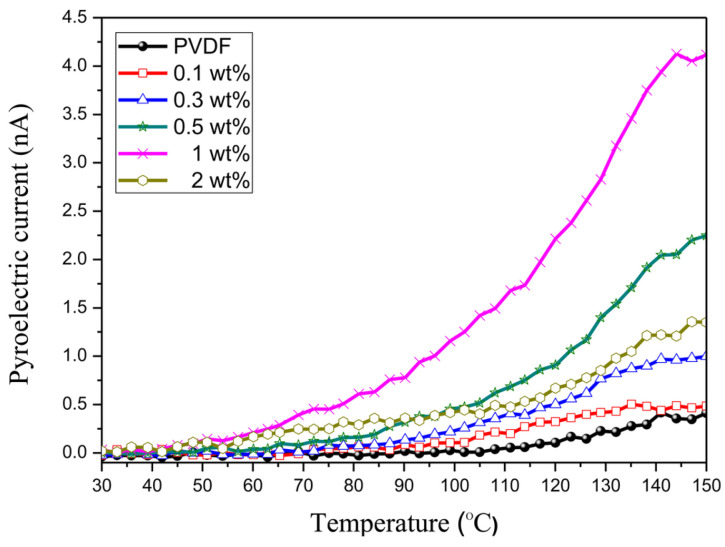
The pyroelectric currents of PVDF film with various wt% GO-doping [122].

**Table 1 micromachines-16-00386-t001:** List of major properties with parameters of PVDF [77,78].

Properties	Parameters	PVDF Example Values
Physical	Density (g/cm^3^)	1.78
	Material color	Transparent resin without color
	Water absorption%	0.02–0.07
	Refraction index (N_D_)	1.40–1.42
	Oxygen index (%)	42–44
Mechanical	Tensile strength @ 23 °C (MPa)	35–55
	Hardness, Shore D	50–80
	Young’s modulus @ 23 °C (MPa)	1340–2000
	Permittivity of free space (F/m)	ε_0_ = 8.854 × 10^−12^
	Elastic constant N/m^2^	C_11_ = 2.184 × 10^9^
		C_12_ = 0.633 × 10^9^
Piezoelectric	Curie temperature (T_c_)	80 °C
		d_31_ = 20–30
	Strain constant d (10^−12^ C/N)	d_33_ = −30
	Stress constant (10^−3^ mV/N)	g_33_ = 335
	Volume resistivity (Ω-cm)	10^15^–2.0 × 10^14^
Thermal	Melting point °C	177
	Defection temperature °C (261 psi)	114–118
	Flammability	V-O
	Thermal expansion coefficient (μm/m-°C)	10^−4^
	Decomposition temp (°C)	375
	Thermal conductivity (W/m-K)	0.144–0.2
	Glass transition temperature, °C	−35
	Flammability °C	V-O

**Table 2 micromachines-16-00386-t002:** Chemical properties of PVDF [10,12,13,14,15,64,65,66,67,68,69,70,71,72,73,74,75,76,77].

Chemical Properties of PVDF					
Organic Solvent Resistance			Alcohol Resistance		
Chemical	20 °C	50 °C	Chemical	20 °C	50 °C
Acetone	Not	Not	Benzyl alcohol (pure)	Yes	Yes
Chlorobenzene	Yes	Yes	Methanol	Yes	Yes
Benzene	Yes	Limited	Ethanol (30%)	Yes	Yes
Chloroform	Yes	Yes	Methyl alcohol (10%)	Yes	Yes
Diethylene glycol	Yes	-	Phenol (10%)	Yes	Yes
Cyclohexane	Yes	Yes	Methyl alcohol (pure)	Yes	Yes
Dimethyl formamide	-	-	Propanol	Yes	Yes
Trichloroethane	Yes	Yes	Phenol (100%)	Yes	Yes
Diethylamino	Yes	Not	Resistance to acids and bases		
Xylol	Yes	yes	Formic acid (10%)	Yes	Yes
Food product resistance			Acetic acid (100%)	Yes	Yes
Glucose	Yes	Not	Acetic acid (10%)	Yes	Yes
Milk	Yes	Yes	Hydrochloric acid	Yes	Yes
Olive oil	Yes	Yes	Sulfuric acid (10%)	Yes	Yes
Wine	Yes	Yes	Lactic acid	Yes	Limited
Vinegar	Yes	Limited	Nitric acid (10%)	Yes	Yes
Resistance to oils and fats			Nitric acid (Conc.)	Limited	Limited
Coconut oil	Yes	Yes	Hydrogen peroxide (90%)	Yes	-
Butyl acetate	Yes	Limited	Sulfuric acid (90%)	Yes	Yes
Mineral oils	Yes	Yes	Sulfuric acid (fuming/monohydrate)	Not	Not
Pine oil	Yes	Yes	Trichlorofluoromethane	Yes	Yes
Paraffin oil	Yes	Yes	Tetrahydrofuran	Limited	Not

**Table 3 micromachines-16-00386-t003:** Different fabrication methods along with their advantages, disadvantages, and parameters of PVDF [68].

Method	Advantages	Disadvantages	Common Parameter
Solution casting	▪ Simple and cost-effective process. ▪ Ability to control film thickness. ▪ Good control over morphology and properties.	▫ Solvent selection can be critical. ▫ Limited scalability due to solvent evaporation. ▫ May require post-treatment steps.	Concentration: 10–20 wt% Temperature: 25–80 °C
Hot pressing	▪ Improved crystallinity. ▪ Enhanced mechanical properties. ▪ Reduced porosity. ▪ Faster processing.	▫ Potential for degradation. ▫ Equipment cost. ▫ Limited control over film thickness and uniformity.	-
Electrospinning	▪ High surface area and porosity in the resulting nanofibrous structure. ▪ Versatile for producing various fiber diameters and architectures. ▪ Potential for fiber alignment.	▫ Requires specialized equipment and high-voltage power supply. ▫ Limited control over fiber alignment and morphology. ▫ Difficulty in scaling up.	Voltage: 10–30 kV Flow rate: 0.5–2 mL/h
Spin coating	▪ Uniform and smooth film formation. ▪ Suitable for small-scale and research applications. ▪ Can achieve controlled film thickness.	▫ Limited scalability due to the size of the spinning apparatus. ▫ Solvent evaporation can lead to uneven film formation. ▫ Difficult to achieve thickness control for thicker films.	Speed: 1000–5000 rpm Time: 30–60 s
Solution coating	▪ Versatile and compatible with various substrates. ▪ Allows for precise control over deposition technique. ▪ Suitable for different scales and applications.	▫ May require optimization for achieving uniform film thickness. ▫ Solvent selection and drying conditions can affect film properties. May require additional post-treatment steps.	Concentration: 5–15 wt% Temperature: 25–60 °C
3D printing	▪ Enables fabrication of intricate 3D structures. ▪ Design flexibility and customization. ▪ Reduced material waste compared with traditional methods.	▫ Limited range of printable materials. Limited mechanical properties compared with conventionally processed materials. ▫ May require post-processing steps for improved properties.	Layer height: 0.1–0.4 mm Speed: 10–100 mm/s

**Table 4 micromachines-16-00386-t004:** The PVDF polymer’s crystalline phase transformation methods along with their advantages and disadvantages.

Methods	Advantages	Disadvantages
Stretching [70,162]	▪ Increased crystallinity ▪ Improved alignment of polymer chains ▪ Enhanced piezoelectric and mechanical properties ▪ Highly reproducible	▫ Potential material degradation ▫ Introduction of defects or stress concentration
Annealing treatment [147,148]	▪ Improved molecular alignment ▪ Increased crystallinity ▪ Enhanced thermal and mechanical properties.	▫ Long processing times ▫ Potential for material degradation ▫ Limited control over phase morphology
Poling [9,163]	▪ Enhanced piezoelectric properties. ▪ Improved alignment of dipoles within the polymer ▪ Increased electroactive response ▪ High d33 coefficient and no requirement for one end structure of the material	▫ Risk of electrical breakdown ▫ High-voltage requirement ▫ Potential for sample damage or degradation

**Table 5 micromachines-16-00386-t005:** The list of distinguishing features between PVDF and P(VDF-TrFE).

Features	PVDF	P(VDF-TrFE)
Type	Homopolymer	Copolymer (VDF and TrFE)
Processing	Demands comprehensive processing to get superior piezoelectric characteristics	Facilitates processing and attainment of desired qualities
Structure	Repetitive units of -CH2-CF2-	Arbitrary integration of VDF and TrFE monomers
Crystallization	May occur in many crystalline phases (α, β, γ)	Crystallizes immediately into the extremely piezoelectric β-phase
Piezoelectricity	Shows piezoelectricity, although needs specialized treatment (e.g., stretching and poling) to further enhance it	Considerably improved piezoelectric properties compared with PVDF
Ferroelectricity	Poor ferroelectric characteristics	Shows significant ferroelectric characteristics

**Table 6 micromachines-16-00386-t006:** Advantages and applications of PVDF copolymers.

Polymer	Advantage	Applications
PVDF	▪ High degree of crystallinity ▪ Excellent mechanical properties ▪ Chemical resistance	▪ Sensor actuators ▪ Energy generation ▪ Energy storage ▪ Microfluids ▪ Biomedical
Poly (VDF-co-TrFE)	▪ Directly crystallizes from melt into electroactive phases ▪ Modified crystallinity ▪ High piezoelectric constant ▪ Thermal hysteresis, high electric output ▪ Sensitivity, flexibility	▪ Sensors actuators ▪ Energy generation ▪ Energy storage ▪ Environmental monitoring and remediation ▪ Microfluids ▪ Biomedical applications
Poly (VDF-co-HFP)	▪ Chemically inert ▪ lower crystallinity due to bulky CF3 groups ▪ High d_31_ piezoelectric constant	▪ Energy storage ▪ Environmental monitoring and remediation
Poly (VDF-co-CTFE)	▪ Optimized piezoelectric properties ▪ High electric-energy density. ▪ High electrostrictive strain response ▪ High dielectric constant ▪ Broader ferroelectric hysteresis loops	▪ Energy storage

**Table 7 micromachines-16-00386-t007:** Advantages and applications of PVDF blends with polymers.

PVDF Blend with Polymers	Advantage	Applications	Refs.
PVDF/PMMA	Promotes the formation of the β-phase	Optical applications Pyroelectric application As a coating	[198,199]
PLLA/PVDF	Improved biodegradability	Energy-harvesting devices	[200,201]
PVDF/ILs	-	Biomedicine to energy storage	[202,203]

**Table 8 micromachines-16-00386-t008:** The characterizations methods for various properties of PVDF.

Properties	Characterization Methods	Determining Parameters
Thermal characterization	Differential scanning calorimetry (DSC)	Melting point, glass transition temperature, crystallization temperature
	Thermogravimetric analysis (TGA)	Thermal stability, degradation behavior
Structural characterization	X-ray diffraction (XRD)	Crystal phases (α, β, γ), crystallinity percentage
	Fourier transform infrared spectroscopy (FTIR)	Characteristic peaks of various crystal phases
Mechanical characterization	Universal testing machine (UTM)	Tensile strength, Young’s modulus, elongation at break
	Dynamic mechanical analysis (DMA)	Viscoelastic behavior
Electrical characterization	Dielectric spectroscopy	Dielectric constant, loss tangent over frequency
	Conductivity measurements	Electrical conductivity
Dielectric	Impedance analyzer	Dielectric constant, loss tangent as a function of frequency
Ferroelectric characterization	Ferroelectric loop tracer	Polarization-electric field (P-E) hysteresis loops
	Switching spectroscopy	Domain switching dynamics
Electromechanical characterization	Strain gauge	Strain generated by electric field
	Laser interferometry	Displacement due to electric field
Piezoelectric characterization	d_33_ meter	Piezoelectric charge coefficient (*d*_33_)
	Piezoresponse force microscopy (PFM)	Piezoelectric domains, piezoelectric coefficients
Pyroelectric characterization	Pyroelectric current meter	Pyroelectric current caused by temperature variation
	Electrocaloric effect measurements	Temperature change induced by electric field

## Data Availability

In this review paper, no new data were created.

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
