# Peer review of "A Comprehensive Review of Piezoelectric PVDF Polymer Fabrications and Characteristics"

_micromachines, 2025, doi:10.3390/mi16040386_

Round 1
Reviewer 1 Report
Comments and Suggestions for Authors
This manuscript gives a comprehensive review on the PVDF organic piezoelectric materials. The manuscrpt is terriblely written. My comments are list here.
1. Some basic knowledges and concepts on ferroelectric and piezoelectric effect such as Fig.3, 4, 5, 6 and Eqs.(1) to (17) can be easily found in some textbook on Ferroelectricity. I suggest the authors delete these content to save time of audiences.
2. Some simple process such as hot pressing of Fig.11 can be clearly describe through text and the Figure 11 should be deleted. The same problem can be also found in Fig. 8, 9,10, 11,12. Please delete these equipment pictures in Fig. 30,31,34,36,37,38,42,44,45,47, 48,49,50,51,52,54,55
3. It's better to give a detail description on the relationship of P(VDF-TrFE) and PVDF.
Author Response
A Comprehensive Review of Piezoelectric PVDF Polymer Fabrications and characteristics
Nadia Ahbab , Sidra Naz , Tian-Bing Xu * , Shihai Zhang
Response to Reviewer 1
General comment:
This manuscript gives a comprehensive review on the PVDF organic piezoelectric materials. The manuscript is terribly written.
Response:
The authors would like to thank the reviewer for his/her valuable, detailed and constructive comments that allow us to further improve the quality of the submitted manuscript. All the comments and suggestions are carefully addressed in this document and the manuscript is accordingly revised. Please find the detailed responses below and the corresponding revisions/corrections highlighted in read of tracked changes in the re-submitted files. In addition, approximately 700 sentences/phrases are revised as highlighted red to make manuscript more readable.
Comment 1:
Some basic knowledges and concepts on ferroelectric and piezoelectric effect such as Fig.3, 4, 5, 6 and Eqs. (1) to (17) can be easily found in some textbook on Ferroelectricity. I suggest the authors delete this content to save time of audiences.
Comment 2: Some simple process such as hot pressing of Fig.11 can be clearly describe through text and the Figure 11 should be deleted. The same problem can be also found in Fig. 8, 9,10, 11,12. Please delete these equipment pictures in Fig.30,31,34,36,37,38,42,44,45,47, 48,49,50,51,52,54,55
Response to comments 1 and 2:
The primary objective of presenting this essential knowledge is to provide a foundational understanding for all kinds of readers in the micromachine community and beyond, particularly three group of readers: micromachines designers, piezoelectric polymer researchers (material engineers, material characterizations), and educational purpose (students, professors from different backgrounds/fields). It is also observed that the basic PVDFs studies performed few decades ago, but its hard-to-find information of PVDF material properties, processing techniques and characterizations in open resource. In addition, there is a gap of PVDF materials development in recent certain decades as the scientists and chemists make material but failed to provide enough details for readers. The second objective of this paper is to provides comprehensive review, to the young generation researchers, on characterization methods and equipment’s regarding to material development, manufacturing processes, copolymerization and characterization for ferroelectric PVDF basic materials and devices. Furthermore, the aim of providing basic equipment is that the reader gets familiar and ideas about available advance equipment’s for characterization of PVDF.
Comment 3:
It's better to give a detail description on the relationship of P(VDF-TrFE) and PVDF.
Response
Thank you so much for this valuable suggestion.
We added detail description on the relationship of P(VDF-TrFE) and PVDF Lines 750-760.
P(VDF-TrFE) is an upgraded variant of PVDF that exhibits markedly improved piezoelectric and ferroelectric characteristics. The unstretched and unpoled PVDF-TrFE exhibited a strong piezoelectric effect, which significantly reduced the device fabrication cost for sensing applications. For example, PVDF-TrFE can be directly spin-coated onto a silicon wafer to form sensors. However, pure PVDF cannot form sensors directly without stretching and poling. The piezoelectric constants of PVDF-TrFE were significantly higher than PVDF films under the same stretching and poling conditions. These enhancements provide it a more adaptable and useful material for many applications, especially those dependent on robust electromechanical coupling.
Table 5. The list of distinguishing features between PVDF and P(VDF-TrFE)
Features PVDF P(VDF-TrFE)
Type Homopolymer Copolymer (VDF and TrFE)
Processing Demands comprehensive processing to get superior piezoelectric characteristics. Facilitates processing and attainment of desired qualities.
Structure Repetitive units of -CH2-CF2- Arbitrary integration of VDF and TrFE monomers
Crystallization May occur in many crystalline phases
(α, β, γ) Crystallizes immediately into the extremely piezoelectric β-phase.
Piezoelectricity Shows piezoelectricity, although needs specialized treatment (e.g., stretching, poling) to further enhance it. Considerably improved piezoelectric properties compared to PVDF.
Ferroelectricity Poor ferroelectric characteristics Shows significant ferroelectric characteristics

Reviewer 2 Report
Comments and Suggestions for Authors
The authors reviewed the characteristics and manufacturing methods of PVDF in this manuscript. However, the manuscript contains numerous typographical and grammatical errors, making it very difficult to read. Additionally, the content is merely a listing of previous research findings, failing to provide readers with any new insights or inspiration.
Author Response
General comment:
The authors reviewed the characteristics and manufacturing methods of PVDF in this manuscript. However, the manuscript contains numerous typographical and grammatical errors, making it
very difficult to read. Additionally, the content is merely a listing of previous research findings, failing to provide readers with any new insights or inspiration.
Response:
Thank you so much for providing precious recommendations and pointing out the grammatical and typo ambiguities. We have updated the manuscript by careful, critical and exhaustive review of the whole draft to improve the language quality by avoiding the grammatical/topographical error, ambiguous sentences and punctuations mistakes for better understanding of the readers as suggested by the worthy anonymous reviewer. Overall, approximately 700 sentences/phrases are revised as highlighted read.
In addition, we provide some insights as following:
I. In introduction, we added:
1) In the last two decades, motivated by the U.S. nano initiative, many researchers have made great efforts to develop various PVDF-based nano-composite materials. (lines 77-78)
2) Although many review papers on PVDF-based materials have been published [2,7], [60–66], there is a gap in the comprehensive review of i) various material proper-ties for device engineering and developers, ii) various characterization methods for materials scientists, engineers and physicists, and iii) material property modifications for other researchers. (line 108-112)
II. In material properties, we added
1) The inverse piezoelectric effect serves as the basis for an electromechanical actuator for electro-deformation. Utilizing the reaction of the material to an applied voltage to provide precise control has been exploited in various piezoelectric actuators[102], [103], [104]. (line 316-319)
III. In fabrication/preparation methods, we added
1) PVDF films are the most popular structures for use in various device applications because, i) films easily form piezoelectric status with relatively lower force stretching to form β-phase and lower voltage to pole to permanent polarization; ii) films make functional devices for sensors, actuators, transducers, and energy harvesting with a) more efficiency, b) lower cost of operations, and lower cost of fabrications. (line 409-413)
2) Massive high-quality commercial piezoelectric films are fabricated with a hot-pressing-based comprehensive production line, which is integrated with hot pressing to form a film, mechanical stretching to transfer to the high-yield -phase, and post-annealing to increase crystallinity. The large-scale commercial production lines, it is relatively easy to control the temperature and stretching uniformity to obtain high-quality piezoelectric PVDF films. However, it is very challenging to obtain high-quality films using the hot-pressing method in a small lab scale. Therefore, the solution casting method is more convenient for use in academic laboratories. (lines 499-506)
3) P(VDF-TrFE) is an upgraded variant of PVDF that exhibits markedly improved piezoelectric and ferroelectric characteristics. The unstretched and unpoled PVDF-TrFE exhibited a strong piezoelectric effect, which significantly reduced the device fabrication cost for sensing applications. For example, PVDF-TrFE can be directly spin-coated onto a silicon wafer to form sensors. However, pure PVDF cannot form sensors directly without stretching and poling. The piezoelectric constants of PVDF-TrFE were significantly higher than PVDF films under the same stretching and poling conditions. These enhancements provide it a more adaptable and useful material for many applications, especially those dependent on robust electromechanical coupling.
Table 5. The list of distinguishing features between PVDF and P(VDF-TrFE)
Features PVDF P(VDF-TrFE)
Type Homopolymer Copolymer (VDF and TrFE)
Processing Demands comprehensive processing to get superior piezoelectric characteristics. Facilitates processing and attainment of desired qualities.
Structure Repetitive units of -CH2-CF2- Arbitrary integration of VDF and TrFE monomers
Crystallization May occur in many crystalline phases
(α, β, γ) Crystallizes immediately into the extremely piezoelectric β-phase.
Piezoelectricity Shows piezoelectricity, although needs specialized treatment (e.g., stretching, poling) to further enhance it. Considerably improved piezoelectric properties compared to PVDF.
Ferroelectricity Poor ferroelectric characteristics Shows significant ferroelectric characteristics
(lines 749-760)
4) Although PVDF composites exhibit multifunctional properties, many challenges must be addressed for practical applications, such as high dielectric loss and instability, strong temperature dependence, and functional separations from multifunction for desired device applications. These disadvantages might be due to i) incorporations be-tween organic PVDF and inorganic fillers, ii) interface space charges between PVDF and fillers, etc. These disadvantages might not be fully reported in the literatures be-cause positive phenomena are much easier to publish than are negative phenomena. (lines 823-829)
IV. In Characterizations, we added
1) Polarity switching measurements were then employed to determine the overall polarization of the material. Raman spectroscopy has recently been used for crystal structure. The mechanical properties can be measured through various tensile tests, the electrical properties of PVDF films can be characterized through electrical impedance spectra with impedance analyzers and others, the electromechanical properties can be achieved through a polarization electrical filed loop (P-E loop), piezoelectric co-efficient, electromechanical coupling factors, etc., and the thermal electrical properties can be obtained by measuring the pyroelectric constants. (Lines 834-842)
2) The dielectric constant measures the potential polarizations of PVDF to estimate the electrical energy generation and storage capabilities. LCR meters or LCR-based impedance analyzers are … was used to study the dielectric properties, such as permittivity and dielectric loss, of PVDF. These properties are fundamental for understanding the ferroelectric and piezoelectric properties of PVDF. (lines 957-964)
3) In addition, impedance analyzer is a basic tool for piezoelectric device engineers. The impedance of a piezoelectric device should be monitored using an impedance analyzer throughout the device fabrication process. (Lines 975-977)
4) The P-E loop is a key parameter for measure the ferroelectric properties of a ferro-electric material. Pr provides the piezoelectric properties of the piezoelectric material. (lines 1007-1008)
5) The resonance method is not commonly used by in the material scientist community. However, it is a convenient method for ferroelectric device engineers to evaluate piezoelectric PVDF films, which can be used for device fabrication. For a piezoelectric-state PVDF film, for instance, a stretched and poled PVDF film, an impedance analyzer can carry the measurement. However, for the PVDF film without stretching and poling, a DC bias field must be applied to convert the PVDF film to the piezoelectric state [227]. (lines 1147-1152)
V. In Summary and future perspectives, we added
1) Films are the most popular shapes used for PVDF materials. Solution casting and coating are the most popular methods for fabricating PVDF films in lab-scale studies. Spin coating provides more uniform PVDF films for micro electromechanical system applications integrated with silicon wafer-based electronics. The hot-press method provides a relatively high-quality piezoelectric film for large-scale commercial fabrication. However, the hot-press method makes it difficult to fabricate unformed films in small lab-scale fabrication. For pure PVDF films, the post-film stretching process to transfer the -phase to the -phase, annealing to increase crystallinity, and electrical poling to form permanent polarization, are needed to improve the ferroelectric and piezoelectric properties. In film processing, i) DSC is the basic method to check the melting, glass transition and crystallization temperatures to determine the annealing conditions, while the thermal stability can be measured with TGA; ii) WAXD, XRD and FTIR are used to determine the -phase; iii) impedance analyzer for dielectric measurements and P-E loop for polarization measurement to determine the electrical, dielectric , and ferroelectric properties of the film; iv) the mechanical properties can be measured with UTM and DMA; v) the direct piezoelectric effect can be determined with d33 meters, PFM, Quasi-Static and dynamic, and acoustic excitation measurements; vi) the inverse piezoelectric effects can be measured by various electrical field-induced strain measurement methods; vii) the electromechanical properties can be determined by the ratio of output mechanical energy over input electrical energy or output electrical energy over the input mechanical energy. In addition, the major electromechanical properties can be measured using impedance resonance method. (lines 1212-1231)
2) Currently, piezoelectric PVDF films are still dominate in practical applications because they are well-studied and mature. (lines 1232-1233)
3) Many PVDF composites exhibit multifunctional properties; however, many challenges have not been addressed for practical applications, such as high dielectric loss and instability, strong temperature dependence, and functional separation from multiple functions for desired device applications. Many disadvantages might not be fully reported in the literature because positive phenomena are much easier be publish than are negative phenomena. More detailed studies are needed in the near future. (lines 1239-1244)

Round 2
Reviewer 1 Report
Comments and Suggestions for Authors
The authors didn't address my previous comments. There are a big content text in this manuscript can be found in some textbook on ferroelectriciy. I can not agree to accept this manuscript for publication.
My previous comments are list here again.
Comment 1:
Some basic knowledges and concepts on ferroelectric and piezoelectric effect such as Fig.3, 4, 5, 6 and Eqs. (1) to (17) can be easily found in some textbook on Ferroelectricity. I suggest the authors delete this content to save time of audiences.
Comment 2: Some simple process such as hot pressing of Fig.11 can be clearly describe through text and the Figure 11 should be deleted. The same problem can be also found in Fig. 8, 9,10, 11,12. Please delete these equipment pictures in Fig.30,31,34,36,37,38,42,44,45,47, 48,49,50,51,52,54,55
Author Response
The authors respect the reviewer’s strong opinion and the suggestions. We deleted all the reviewer’ requested items and renumbered them as seem as this text color in this version. Please check.

Reviewer 2 Report
Comments and Suggestions for Authors
The manuscript has been well revised. However, it still contains some errors. Refer to the highlighted sections in the attached file. Additionally, here are parts where the journal names in the references are not displayed in abbreviations, so revise them according to the guidelines.

Overall, English expression is good.
Author Response
The authors thank the reviewer’s carefully review and catches. All the highlights are corrected in the color as seem as this text color. We also revised the references according to the guidelines.
